# A tracer study for the development of in-water monitoring, reporting, and verification (MRV) of ship-based ocean alkalinity enhancement

Adam V. Subhas[1], Jennie E. Rheuban[1], Zhaohui Aleck Wang[1], Daniel C. McCorkle[2], Anna P. M. Michel[3], Lukas Marx[1], Chloe L. Dean[1,4], Kate Morkeski[1], Matthew G. Hayden[1], Mary Burkitt-Gray[3], Francis Elder[3], Yiming Guo[1], Heather H. Kim[1], Ke Chen[5]

[1]Department of Marine Chemistry and Geochemistry, Woods Hole Oceanographic Institution, Woods Hole, MA, USA
[2]Department of Geology and Geophysics, Woods Hole Oceanographic Institution, Woods Hole, MA, USA
[3]Department of Applied Ocean Physics and Engineering, Woods Hole Oceanographic Institution, Woods Hole, MA, USA
[4]MIT-WHOI Joint Program in Oceanography, Massachusetts Institute of Technology, Cambridge, MA, USA
[5]Department of Physical Oceanography, Woods Hole Oceanographic Institution, Woods Hole, MA, USA

*Correspondence to*: Adam V. Subhas (asubhas@whoi.edu)

**Abstract.** Ocean alkalinity enhancement (OAE) is a marine carbon dioxide removal (mCDR) approach that relies on the addition of liquid or solid alkalinity into seawater to take up and neutralize carbon dioxide ($CO_2$) from the atmosphere. Documenting the effectiveness of OAE for carbon removal requires research and development of measurement, reporting, and verification (MRV) frameworks. Specifically, direct observations of carbon uptake via OAE will be critical to constrain the total carbon dioxide removal (CDR), and to validate the model-based MRV approaches currently in use. In September 2023, we conducted a ship-based rhodamine water tracer (RT) release in United States federal waters south of Martha's Vineyard, MA followed by a 36-hour tracking and monitoring campaign. We collected RT fluorescence data and a suite of physical and chemical parameters at the sea surface and through the upper water column using the ship's underway system, a CTD rosette, and Lagrangian drifters. We developed an OAE analytical framework that explicitly references the OAE intervention and the resulting CDR to the baseline ocean state using these *in situ* observations. We evaluated the effectiveness of defining a "dynamic" baseline, in which the carbonate chemistry was continuously constrained spatially and temporally using the shipboard data outside of the tracer patch. This approach reduced the influence of baseline variability by 25% for $CO_2$ fugacity ($f$CO$_2$) and 60% for TA. We then constructed a hypothetical alkalinity release experiment using RT as a proxy for OAE. With appropriate sampling, and with suitable ocean conditions, OAE signals were predicted to be detectable in total alkalinity (TA >10 umol kg$^{-1}$), pH (>0.01) and $CO_2$ fugacity ($f$CO$_2$ >10 μatm). Over 36 hours, an ensuing additional $CO_2$ uptake was driven by this persistent gradient in surface $f$CO$_2$. The calculated CDR signal was detectable as a 4 μatm surface $f$CO$_2$ increase, a pH decrease of 0.004 units, and a dissolved inorganic carbon (DIC) increase of 1.8 μmol kg$^{-1}$, translating to 10% of the total potential CDR. This signal, and the CDR itself, would continue to grow as long as an $f$CO$_2$ gradient persisted at the sea surface. Climatological results from a regional physical circulation model supported these findings and indicated that models and in-water measurements can be used in concert to develop a comprehensive MRV framework for OAE-based mCDR.

# 1 Introduction

Carbon dioxide ($CO_2$) emissions reductions and a transition to non-fossil fuel energy are essential for mitigating the worst effects of climate change, but there is mounting evidence that emissions reductions alone will not be sufficient to do so (IPCC AR6; NRC, 2015). The internationally recognized target of limiting mean warming to below 2°C will require supplementing large-scale emissions reductions with $CO_2$ removal from the atmosphere, to deal with legacy emissions and to neutralize residual emissions

from hard-to-abate sectors (Lamb et al., 2024). The oceans are the largest carbon reservoir on Earth's surface, and attention from the private sector, academia, and federal agencies, is being focused on evaluating and deploying marine carbon dioxide removal (mCDR) strategies to help meet this climate goal (NASEM, 2021).

One mCDR approach, Ocean Alkalinity Enhancement (OAE), encompasses a suite of processes

involving the intentional addition of alkaline materials to seawater to increase its buffering capacity, driving an enhanced uptake of $CO_2$ from the atmosphere (Renforth and Henderson, 2017). The private sector is already deploying OAE technologies (Kitidis et al., 2024), funded through a growing voluntary carbon market. Recently, researchers have come together to establish best practices for OAE research and development (Oschiles et al., 2023). Of specific interest is the research required to establish

measurement, reporting, and verification (MRV) frameworks for OAE (Ho et al., 2023). Such frameworks require the attribution of a CDR signal to an OAE intervention (i.e. additionality), and must reliably attribute OAE contributions to net $CO_2$ removal, over and above background carbon fluxes.

Research on open-water OAE deployment and its associated MRV is currently lacking, and in-water experiments are critical for advancing the field (Cyronak et al., 2023). Currently, the only

monitoring framework for OAE exists as part of a commercial wastewater outfall MRV protocol (Isometric, 2024). This protocol focuses on in-water measurements not for the measurement of CDR, but instead to verify dispersal limits in the near-field of the outfall pipe, and for the calibration of a model which is then used for CDR calculations. Model-based approaches will be a critical part of MRV, given the large spatial scales and long open-ocean $CO_2$ uptake timescales relevant for climate-scale

OAE (Zhou et al., 2024). However, direct measurements of the oceanic carbon sink are essential for providing independent estimates on critical carbon cycle properties, and for continued model validation and groundtruthing as anthropogenic and natural carbon sinks change through time (Friedlingstein et al., 2024). It is therefore critical to evaluate the conditions under which in-water measurements can be used directly for CDR quantification, and to develop a framework for utilizing these measurements for MRV.

Open-water experiments will help to establish ways to account for temporal and spatial variability, to determine signal-to-noise and detection limits for OAE and associated $CO_2$ uptake, and to validate models that can be used to extrapolate OAE signals in space and time once the initial enhancements become indistinguishable from the baseline (He and Tyka, 2022). These factors will feed into the establishment of in-water MRV frameworks, and will likely need to be specific to the method

of OAE deployment (e.g. wastewater outfall, ship-based, or sediment-based; Eisaman et al., 2023). Moreover, these in-water tests can aid in evaluating near-field models of dispersion and dilution, with implications for the practical deployment of alkalinity in seawater and its associated MRV.

Prior to open-water alkalinity dispersal experiments, tracer-based studies can be used to evaluate the physical dispersion of water masses, to inform MRV frameworks, and to anticipate the potential

outcome of OAE deployments. The development of MRV benefits from the construction of research-grade analytical frameworks that can then be adapted for practical, scalable applications. For example, the temporal and spatial baseline variability can be assessed, as well as practical aspects of MRV, including methodologies for accurately sampling OAE interventions and background values for assigning CDR additionality. To this end, we conducted a tracer study in September 2023 in which we dispersed rhodamine water tracer dye (RT), followed by an intensive monitoring campaign using a research vessel and Lagrangian drifters. The campaign is named LOC-01, the first field campaign of the Locking Ocean Carbon in the Northeast Shelf and Slope (LOC-NESS) Project. We compare the resulting datasets with ship wake dilution models to refine dispersal strategies, and assess the effect of baseline variability on the carbonate system. We propose an MRV framework for ship-based, liquid alkalinity OAE approaches, although this framework may also be generalizable to other forms of OAE. We use this framework to simulate an OAE dispersal experiment and evaluate the potential for detecting OAE signals over and above real-world baseline variability. We conclude with recommendations for future in-water OAE dispersal and monitoring experiments.

## 2 Methods

### 2.1 Proposed Analytical Framework for OAE and its CDR

Constructing an analytical framework for mCDR, and for OAE specifically, requires defining the main processes at work. Here we define three main steps in our analytical framework:

1. Net Alkalinity transfer from alkaline feedstock into seawater via dissolution;
2. Tracking of dissolved alkalinity and its dispersion (and for solid feedstocks, particle transport and settling);
3. Calculation of CDR due to the above processes, via direct measurement, models, and/or a combination of both.

Here, we assume a liquid form of alkalinity e.g. sodium hydroxide, such that in Step 1 alkalinity transfer efficiency is very high and can be restricted to the sea surface. In Step 2, we solely focus on dissolved alkalinity tracking.

For all steps, a baseline state must be established. The carbon removal in Step 3 must be attributable to the intervention (i.e. the alkalinity enhancement), in this case driven by a reduction in surface water $fCO_2$ (where $fCO_2$ is the fugacity, or effective partial pressure, of $CO_2$ in seawater), and the resulting $CO_2$ uptake into the surface ocean. In ocean model-based MRV, baselines are defined by a "control" run without any mCDR (Isometric 2024, He and Tyka, 2022). In-water MRV frameworks, on the other hand, require careful consideration of baselines, which could be established from historical data or from in-water data sampled at the same temporal and spatial resolution as the intervention itself. In the following analytical framework, we explicitly distinguish between OAE-driven signals (Steps 1 and 2) and CDR-driven signals (Step 3):

$$TA_t = TA_{bl,t} + \Delta TA_{OAE,t} + \Delta TA_{CDR,t}; \qquad \text{(eq. 1a)}$$
$$DIC_t = DIC_{bl,t} + \Delta DIC_{OAE,t} + \Delta DIC_{CDR,t}; \qquad \text{(eq. 1b)}$$
$$pH_t = pH_{bl,t} + \Delta pH_{OAE,t} + \Delta pH_{CDR,t}; \qquad \text{(eq. 1c)}$$
$$fCO_{2\,t} = fCO_{2\,bl,t} + \Delta fCO_{2\,OAE,t} + \Delta fCO_{2\,CDR,t}. \qquad \text{(eq. 1d)},$$

Total alkalinity (TA), dissolved inorganic carbon (DIC), pH, and the fugacity of $CO_2$ in seawater ($fCO_2$) are functions of time $t$. We separate out measured carbonate chemistry parameters across the lifetime of an OAE deployment (the left-hand side of eqs. 1a-d) into three components on the right-hand side: 1) The baseline (subscript bl); 2) the change in the parameter due to OAE (subscript OAE); and 3) the change due to the subsequent CDR (subscript CDR). We include biogeochemical feedbacks to the alkalinity addition into $\Delta TA_{OAE,t}$ (eq. 1a) such that it represents the net change in TA as a result of the OAE intervention. For example, in this framework, mineral precipitation reactions consuming alkalinity (Moras et al., 2022, Hartmann et al., 2022), or changes in natural alkalinity cycling (Bach, 2023, Lehmann and Bach, 2025) would be included in this term as it represents the *net* addition of alkalinity responsible for CDR. Because $CO_2$ uptake does not affect TA, $\Delta TA_{CDR,t}$ is by definition zero. For completeness, we leave the term shown in the equation.

We assume no biological feedbacks on DIC, meaning that $\Delta DIC_{OAE,t}$ is zero (eq. 1b). We justify this assumption from the recent literature suggesting that modest TA enhancement does not significantly impact phytoplankton primary production, net community production, or zooplankton and fish development in multiple locations (Subhas et al., 2022; Ferderer et al., 2022; Camatti et al., 2024; Bednarsek et al., 2024; Goldenberg et al., 2024). Similarly, we do not consider mineral precipitation feedbacks on OAE, assuming that the dispersal conditions would be controlled to limit and/or avoid this process in the near-scale. Furthermore, we ignore trace amounts of $CO_2$ absorbed by the feedstock before dispersal, and/or trace amounts of carbonates that could produce DIC upon dissolution. Future work should be conducted to demonstrate whether such feedstock impurities, and biological and geochemical feedbacks, need to be incorporated into net TA and DIC terms associated with OAE and its CDR. We explicitly define OAE and CDR terms for nonconservative carbonate system parameters pH (eq. 1c) and $fCO_2$ (eq. 1d) as well, as these measurements are routinely collected *in situ* for and thus will be critical for MRV.

To illustrate the usage of this framework, we show a pulsed addition of pure, liquid alkalinity to the surface ocean, followed by a dilution with surrounding seawater back to the baseline (Fig. 1). This example assumes the case where the size of the intervention is small relative to the volume of the water body, such that infinite dilution can be assumed. While we have constructed this schematic for pure alkalinity (i.e. alkalinity generated from non-carbonate mineral, liquid feedstocks), a similar construction could be used for carbonate-based alkalinity where DIC is added along with TA. The system would become slightly more complex, but equally tractable. In addition, solid feedstock dissolution rate and sinking velocity would need to be considered, and would spatially and temporally decouple alkalinity generation and dispersal from $CO_2$ uptake at the sea surface. The blue line in Figure 1 indicates changes in each parameter due to the OAE intervention. For pure alkalinity addition to the sea surface, there is a step change in TA, but no corresponding step change in DIC (i.e., no blue line in

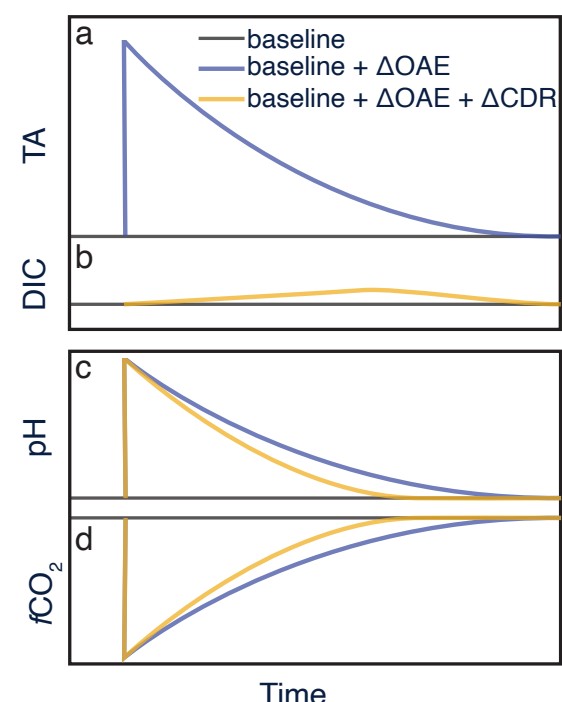

**Figure 1:** Schematic of a pulsed pure alkalinity addition to seawater. Perturbations related to OAE ($\Delta$OAE) are shown in blue traces. Perturbations related to the resulting $CO_2$ uptake ($\Delta$CDR) are shown in yellow lines. The difference between these two curves is the true CDR signal. **a)** TA responds to OAE, and decays away with time. **b)** DIC responds to gas exchange, representing the CDR response to the OAE intervention. **c)** pH and **d)** $fCO_2$ respond to both OAE and CDR perturbations.

Fig. 1b). The alkalinity pulse results in an immediate increase pH (Fig. 1c) and drop in $fCO_2$ (Fig. 1d), followed again by dilution and a return to baseline conditions. It is the air-sea gradient in $fCO_2$ ($\Delta fCO_2$) that drives the subsequent uptake of atmospheric $CO_2$, which constitutes the CDR process (yellow lines, Figs. 1b, c, d).

The resulting OAE+CDR signal creates a small residual that must be distinguished from both the OAE-only signal, and the baseline, in order to document that CDR has occurred. The ingrowth of this signal is slow due to the sluggish exchange kinetics of $CO_2$ between the surface ocean and the atmosphere (Jones et al., 2014). No change occurs in TA as a result of $CO_2$ uptake, such that TA solely responds to the OAE forcing (i.e. no yellow line in Fig. 1a, equivalent to setting $\Delta TA_{CDR,t} = 0$). In contrast, DIC only responds to CDR, with the signal growing in slowly over time and then dissipating back to baseline conditions (Fig. 1b). Non-conservative carbonate system parameters such as pH and $fCO_2$ (as well as others, i.e. saturation state) will always respond to both OAE and CDR signals. Maintaining a sustained, measurable gradient in $fCO_2$ is therefore central to observing a CDR signal, and will be dependent on physical, chemical, and biological processes. For example, the dispersion and dilution of water masses, both horizontally and vertically, will be a critical factor, as will the ability to track the intervention through space and time. This is the central challenge of in-water MRV (Ho et al., 2023).

We note that this framework is unique to open-water alkalinity additions, and may not hold true in all environments, especially sedimentary alkalinity

additions where extensive pore fluid exchanges and reactions can modify the production and consumption of alkalinity *in situ* (Bach, 2023). Furthermore, alkalinity additions could start to alter alkalinity cycling processes in the open ocean as well, such as the formation and dissolution of biogenic $CaCO_3$ in the euphotic and mesopelagic zones (Subhas et al., 2022, Ziveri et al., 2023, Dean et al., 2024). Such considerations would need to be built into the

framework as modifications to the baseline biogeochemical ocean state, or as a modification to the $\Delta TA_{OAE}$ or $\Delta DIC_{OAE}$ signals (Bach, 2023, Lehmann and Bach, 2025).

## 2.2 Research Plan

The dye release and monitoring operations were conducted from the *R/V Connecticut*, a 27.4 meter (90-foot) research vessel operated by the University of Connecticut (UConn). The experiment was carried out in federal waters south of Martha's Vineyard, guided by previous studies in the region demonstrating the effectiveness of plume tracking using rhodamine water tracer (RT) dye over hours to days (Rypina et al., 2021, Proehl et al., 2005), as well as the effective pairing of rhodamine with alkalinity releases over short timescales (Albright et al., 2016, Cyronak et al., 2023). Originally scheduled for August 24th, we decided to postpone the experiment by one week, finding a ~3 day window characterized by low winds (gusts less than 6 m s$^{-1}$), low swell and waves (less than about 1 m) and cloudless skies, starting on September 1st. We departed from Avery Point, CT on the night of September 1st, with onsite operations commencing on the morning of September 2nd.

Using RT as a tracer offers several advantages compared to other water tracers. RT fluorometers are relatively inexpensive (~$2,000-10,000 USD), widely available, and can be mounted on a range of oceanographic platforms, and RT is easily mixed and dispersed into seawater. Sampling resolutions of up to 8 Hz (Busch et al., 2013) and detection limits down to 0.01 ppb can be achieved, depending on the instrument model and environmental conditions (Hixson & Ward, 2022).  Due to its strong pink-red color, RT is visually identifiable at concentrations of ~tens of ppb. Because of its visual properties, it can also be detected using optical techniques on a variety of platforms (e.g. Johansen et al. 2022a, 2022b, Sundermeyer et al., 2007).  The downside to RT is that it is not as sensitive of a tracer as inert dissolved gas tracers (e.g. $SF_6$ or $^3He$) that exhibit higher signal-to-noise ratios and low detection limits in the parts per trillion level. These inert gas tracers are highly insoluble in seawater, allowing for further calculation of tracer losses due to air-sea gas exchange, given some knowledge of the physical mixing and dispersion of the tracer patch (Ho et al., 2011, Doney et al., 2024). However, measurements of these tracers are time- and labor-intensive, often taking minutes to tens of minutes to complete on specialized instrumentation.  Low-power *in situ* instrumentation that could be installed on drifters or small vehicles is also not typically available for analysing inert gas tracers. Although significant uncertainty exists for extending open-ocean gas transfer rates to inshore environments (Long and Nicholson, 2017), these relationships are widely used for open-ocean conditions (Wanninkhof, 2014). Due to the three-day duration of our experiment, the requirement for high-resolution sampling in the dynamic coastal environment of the Northeast Shelf, and the widely used gas transfer characteristics of this setting, we found that RT was both necessary and sufficient for our study.

## 2.3 Rhodamine dye dispersal

A tank of RT dye was prepared by adding 56 kg of powdered rhodamine water tracer dye (Kingscote FWT Red Powder 105403-25lb) to a 1,000 liter (275-gallon) intermediate bulk container (IBC). Approximately 829 liters of fresh water was added via a hose at the University of Connecticut Avery Point dock. An additional 117 liters of isopropanol were added to adjust the final density of the solution to approximately that of surface seawater (1.021 kg m$^{-3}$). This solution was vigorously mixed to ensure complete dissolution of the rhodamine dye powder, resulting in a dark purple, slightly viscous solution with an estimated concentration of 0.0571 kg RT kg$^{-1}$ solution. The IBC was covered with an opaque tarp to prevent photodegradation of the rhodamine dye during storage and transport.

A pre-dispersal site survey and collection of baseline chemical and biological data was carried out before dawn and the dispersal of rhodamine began at daybreak to provide maximum daylight for plume tracking during the first day. Release of the RT dye was accomplished using gravity feed. The 2-inch
ball valve at the outlet of the IBC was fully opened during dispersal and the dye was routed through a 2-inch internal diameter lay flat hose (McMaster-Carr #5295K35). The end of the hose was secured to a polyethylene plate using self-tapping screws. The plate was bridled to the ship to allow the dye mixture to fan out along the plate's surface and enter the seawater with a mostly horizontal trajectory and low velocity. The ship steamed in a spiral pattern during dispersal, starting from a central point and working
outwards (See Results, Fig. 3). The dispersal pattern was established by the ship's captain visually following the outer edge of the dispersal spiral.

Comparison to the ship wake model of Chou (1996) was done by calculating dilution ratios using underway rhodamine signals, compared to the initial IBC concentration of rhodamine (D = $RT_{init}/RT_{underway}$, where $RT_{init} = 0.0571$ kg RT kg$^{-1}$ solution, or $5.71 \times 10^7$ ppb). The dilution model is a
semi-empirical description of ship-wake dilution using the formula:

$$D = 0.2107/Q_eU^{1.552}t^{0.552}B^{1.448}, \text{ (eq. 2)}$$

Where the dilution D is a function of the dispersal rate $Q_e$ (m s$^{-1}$), the ship speed $U$ (3.8 knots, or 1.95 m
s$^{-1}$), time $t$ (s), and vessel width B (m), and is valid out to distances of less than 100B. The *R/V Connecticut* has a beam of 7.9 m, and the material was discharged at approximately 0.2 L s$^{-1}$. The IMCO (Intergovernmental Maritime Consultative Organization) dilution formula, presented by Chou (1996), presents a simpler calculation as a formulation of $Q_e$, U, t, and ship length L, rather than ship width:

$$D_{IMCO} = 0.003/Q_e*U^{1.4}L^{1.6}t^{0.4}. \text{ (eq. 3)}$$

The *R/V Connecticut* is 27.4 m long. Both equations are used to compare to dilution data from the dispersal period below.

**2.4 Monitoring**

The monitoring strategy involved repeated sampling through the patch, starting inside, moving outside, and traveling back again. This approach allowed for baseline (out of patch) and experimental (inside patch) samples paired closely in time and in space, allowing us to assess the additionality of the intervention. Periodically, vertical samples via CTD rosette were taken to assess the vertical distribution
of dye and other water column properties. Because the vertical loss of dye is significantly slower than the horizontal spreading (Rypina et al., 2021), these vertical samples were spaced further out in time, averaging about every four hours. Monitoring consisted of three main approaches: 1) continuous surface water sampling using the ship's underway system, at a frequency of at least every 10 minutes and as fast as every second, depending on the parameter (Section 2.3.1); 2) "in-patch" and "out of patch" CTD
rosette casts both pre-dispersal and roughly every four hours after dispersal, in order to determine the vertical water column structure and tracer distribution; and 3) Lagrangian drifters equipped with GPS

and sensors to follow the patch. We attempted to conduct CTD casts at the highest RT concentrations measured on the underway system, although ship drift meant that we did not hit the peak once the CTD entered the water. Out-of-patch locations were determined visually by reaching near-baseline underway
RT concentrations.

Rhodamine fluorescence measurements were acquired with four Cyclops 7F fluorometers (Turner Designs, #2110-000-R). One Cyclops 7F with a shade cap (Turner Designs, #2100-701) was integrated to the CTD rosette for profile measurements. One Cyclops 7F was connected to the ship's underway system using a flow-through cap (Turner Designs, #2100-600) and logged continuously using a
DataBank (Turner Designs, #2900-010). Two Cyclops 7F fluorometers were integrated into interchangeable PME Cyclops-7 Loggers with shade caps (Precision Measurement Engineering, Inc) for *in situ* data logging and were deployed on the Lagrangian drifters with one-minute measurement intervals. An initial single-point calibration was run prior to the cruise using pre-made 400 ppb Rhodamine WT Dye (Turner Designs, #6500-120) and deionized water at 23 °C. A post-cruise
calibration was run by preparing a 400 ppb solution of the dye used in this field study (Kingscote FWT Red Powder 105403-25lb) dissolved in 0.2-micron-filtered seawater acquired from 300 m offshore in Martha's Vineyard Sound (41.530668, -70.645629) by the Environmental Systems Laboratory (ESL), Woods Hole Oceanographic Institution. For measurements at 100x gain, the lowest recorded value in the field from outside the dye patch was used as the blank (baseline) calibration value to account for
real background fluorescence (e.g., chlorophyll) or turbidity. For measurements at 1x and 10x gain, filtered seawater was used to acquire the blank (baseline) calibration values. For all post-cruise calibrations, the same hardware was installed on each fluorometers as had been used for deployment, e.g., flow-through cap, shade cap, or data logger. Calibration parameters were determined following the equations in the manufacturer's manual (Turner Designs, 2023).

### 2.4.1    Ship Underway System

We used the ship's underway system for real-time plume tracking via high-resolution RT fluorescence measurements. These measurements proved critical for tracking the patch at night and once the RT signal was no longer visible by eye. As described below, the rhodamine fluorometer signal was fed in real-time to a monitor on the bridge to allow for rapid navigation decisions. The ship's
underway system was pumped from 1.5 meters below the sea surface via a Hayward Lifestar series aquatic pump (300 L min$^{-1}$). The wet lab was fed from this pump through a 2-inch (5 cm) schedule 80 gray PVC pipe running approximately 25 ft. (7.6 m) from the intake, corresponding to a travel time from intake to lab of approximately 30 seconds. A split into the lab was fed via a1/2-inch tube. Upon entering the lab, a split from this line was teed off to feed a Contros HydroFIA underway total alkalinity
analyzer and a General Oceanics underway $p$CO$_2$ system. The second arm of the tee connected to a debubbler (~1L volume) with a flow rate of 2-3 L min$^{-1}$, which fed the ship's thermosalinograph (SBE45) and a Turner Cyclops 7F rhodamine fluorometer.

Surface seawater and atmospheric xCO$_2$ (mole fraction of CO$_2$) was continuously measured with the underway $p$CO$_2$ system (Model 8050, General Oceanics, FL, USA) following the best practice of
seawater CO$_2$ measurements (Dickson et al., 2007). Measured xCO$_2$ values were converted to $f$CO$_2$ or $p$CO$_2$ based on Dickson et al. (2007) for reporting and flux calculation. Surface water was pumped to

the $pCO_2$ system via the shipboard underway system, while fresh air samples were pumped continuously by the $pCO_2$ system from the top of the research vessel away from any potential $CO_2$ contamination (e.g., ship exhausts). The system was calibrated every five hours with three $xCO_2$ gas standards traceable to or consistent with the World Meteorological Organization (WMO) standards plus a zero gas. The system was configured for a measurement frequency of every two minutes for surface seawater $xCO_2$ and five hours for atmospheric $xCO_2$, with a precision and accuracy of ~0.2% (e.g., about ±1 µatm at 400 µatm $fCO_2$ level). All $fCO_2$ or $pCO_2$ data were corrected for water vapor by the detector and reported as values in 100% humidity at *in situ* temperature (measured from the ship's underway thermosalinograph).

We installed a CONTROS HyrdoFIA® TA flow-through analyzer (4H-JENA engineering GmbH, Germany) into the *R/V Connecticut* underway seawater supply (following Seelmann et al., 2019; 2020), prior to the de-bubbling system feeding into the thermosalinograph SBE45 (Seabird Scientific). The source water was directed through a Repligen cross-membrane (MiniKros, 0.2 µm PES) filter into a 250 ml sample holding vessel, to accumulate sufficient volume for analysis and to overcome delay in measurements due to instrument analysis time (~ 9.5 min per sample). From this, the CONTROS analyzed TA as single-point open-cell titration with 0.1 M hydrochloric acid (HCl) and subsequent spectrophotometric pH detection with 0.002 M bromocresol green (BCG). Both HCl and BCG were freshly made up prior to the cruise and the CONTROS was calibrated onshore using Dickson seawater (CRM Batch #205). The system was conditioned by running 25 underway seawater measurements during the vessel transit, after which we calibrated the system with 5 measurements of secondary in-house reference seawater, calibrated against Dickson CRM Batch #205. The system was set to continuous measurement mode with sampling intervals of 6 samples per 60 minutes. Over the duration of the cruise, we collected 294 continuous measurements for TA using the CONTROS system. Periodically, every ~2 hours, we collected discrete samples (250 ml) from the cross-membrane filter outflow for cross-calibration, poisoned with 50 µl saturated mercuric chloride solution. These were analyzed on-shore using a Metrohm titrator, consisting of an 805 Dosimat and an 855 Robotic Titrosampler. After deployment, the system was again calibrated with secondary reference seawater for offset and drift correction, resulting in a mean accuracy of 0.31%. Precision of the instrument was better than 3 µmol kg$^{-1}$. Outliers are a known problem with this analyzer (Seelmann et al., 2019). Outliers were identified visually and removed. An alternate approach, in which outliers greater than two standard deviations away from a rolling 6-sample mean, gave similar results to visual inspection, resulting in 8 points (2.7% of all data) removed visually, versus 10 points (3.4% of all data) removed using a standard deviation cutoff (Fig. S1). TA was finally recalculated using *in-situ* temperature and salinity.

After the debubbler and the thermosalinograph, we installed a rhodamine fluorometer with a flow-through cap, connected to the Turner DataBank. The rhodamine fluorometer was set to sample at 0.5 Hz. This sampling frequency optimized limitations on memory storage, battery capacity, and data download time from the Turner DataBank. Because of these limitations, RT fluorescence was measured continuously for approximately four hour stretches, with ~10-minute gaps while the datalogger was downloaded, recharged, and the memory was reset. The fluorescence data, along with the ship's navigation data, were read continuously to a laptop running a data mapping tool using Matlab, which was mirrored to a monitor located on the bridge.

### 2.4.2 CTD rosette sampling

The first CTD rosette casts occurred once on station before the release to collect baseline samples. Subsequent CTD casts were taken after the dispersal, both within and outside of the patch, identified visually and using the ship's underway system. The CTD rosette consisted of twelve 5-L Niskin bottles and a vessel-provided sensor suite including temperature, conductivity, and depth (SBE 03, 04, 02, respectively), pH (SBE18), dissolved oxygen (SBE43), and chlorophyll-a (Wetlabs-wetstar). The science team provided a rhodamine fluorometer (Turner Cyclops 7F) that was integrated into the auxiliary port of the Seabird SBE911Plus CTD.

Bottle samples were taken from the Niskin rosette, with duplicate bottles taken at 20, 16, 12, 10, 5, and 1 meter depths. One Niskin was sampled for dissolved constituents (dissolved inorganic carbon (DIC), total alkalinity (TA), nutrients ($NO_3$+$NO_2$, $NH_4$, $PO_4$, silicate), and rhodamine fluorescence). The second bottle was sampled for particulate material, including PIC, POC, and microbial community abundance via flow cytometry.

Samples for DIC and TA were collected into 250 ml narrow-neck borosilicate bottles by directly filtering through a 0.45 μm filter cartridge. Sampling bottles were rinsed thoroughly and filled from the bottom, overflowing with three times the sample volume, and poisoned with 50 μl saturated mercuric chloride solution (following Dickson et al., 2007). DIC and $\delta^{13}$C-DIC were determined from triplicate analysis via an Apollo AS-D1 in line with a Picarro G-2131i cavity ringdown system (Su et al., 2019), calibrated with in-house secondary seawater standards, intercalibrated against Dickson Certified Reference Materials (Batch #205). Absolute $\delta^{13}$C for seawater standards was calibrated by running seawater against solid reference materials (e.g. IAEA-C2, NBS-19, NBS-18, NBS-20, TIRI-F) on an Automate prep device coupled to the same Picarro G-2131i (Subhas et al., 2015, Subhas et al., 2019). TA was determined for triplicate samples by open-system Gran titration using a Metrohm 805 Dosimat and an 855 Titrosampler with 0.04 M HCl as titrant. TA was determined via a nonlinear least-squares method (following Dickson et al., 2009) and TA analysis was warranted with in-house secondary seawater standards run intermittently as triplicates after 15 individual titrations.

Nutrient samples were collected subsequently into 15 ml Falcon tubes and stored frozen (-20 °C) until onshore analysis at the Woods Hole Oceanographic Institution nutrient analytical facility against certified reference materials (Batch CL-0438, KANSO). Particulate samples were collected by vacuum filtration of 4 L over 0.2 μm pre-combusted (4 h, 500 °C) glass-fiber filters. Filters were stored frozen (-20 °C) in individual pre-combusted aluminum foil envelopes. On land and after drying (60 °C) overnight, filters were cut precisely in half using a sterilized ceramic roller blade. Half a filter was tightly packed in a tin capsule (EA Consumables) and sent to the UC Davis Stable Isotope Facility for analysis of particulate carbon (PC), nitrogen (PN) and $\delta^{13}$C-PC via an elemental analyzer coupled with an isotope ratio mass spectrometer. The second half of the filter was analyzed for particulate inorganic carbon (PIC) on a Picarro-Automate autosampler measuring [$CO_2$] and $\delta^{13}$C-$CO_2$ after converting all $CaCO_3$ to $CO_2$ by acidification with 10% phosphoric acid. Particulate organic carbon (POC) was calculated as the difference between total particulate carbon (PC) and PIC.

### 2.4.3 Lagrangian Drifters

The drifters used to trace the dye patch were based on the Student Built Drifter design developed at the National Oceanic and Atmospheric Administration (NOAA) Northeast Fisheries Science Center in Woods Hole, Massachusetts (Manning et al, 2009). These designs have remained essentially the same since the 1980's and are evolutions on the Davis-style "CODE" (U.S. Coastal Dynamics Experiment) surface drifters first developed at Scripps Institution of Oceanography (Davis, 1985). These designs
comply with the World Ocean Circulation Experiment specifications of 40:1 drag ratios. All the drifters used in this experiment had a 1 m drogue depth. Four drifters were deployed in the dye patch.

        All four drifters were deployed at once, at the same location, to assess how much they would drift from each other, and the patch, over the deployment. Tracking was accomplished using SPOT Trace satellite tracking devices. The SPOT trace reports its position with a 5-minute frequency and an
410 accuracy of approximately 5 m. Two of the drifters had a Turner Cyclops-7F rhodamine fluorometer, logging at 1 minute intervals at a fixed depth of 2.5 m, and an In-Situ Aquatroll 600 multiparameter sondes measuring pH, temperature, conductivity and dissolved oxygen at 1 minute intervals, mounted alongside the fluorometer.

**2.4.4 Satellite Imagery**

        High resolution satellite imagery was collected during the cruise from Planet Labs via two methods: 1) Ultra-high resolution (0.5m per pixel) multiband imagery (red, green, blue, near-infrared, and panchromatic) was collected via the Planet SkySat constellation through tasked image collection and 2) high resolution (3.0m per pixel) multispectral imagery (8-band) through the PlanetScope near-daily
revisit product via the Dove/SuperDove constellation. Three images were collected during the cruise, on September 2, 2023 at 14:14:25 UTC, September 2, 2023 at 18:58:16 UTC, and September 3, 2023 at 14:43:07 UTC. The first two images were collected via SkySat tasking, and the third via PlanetScope. Level 0 images were internally processed by Planet's algorithms for orthorectification and atmospheric correction to produce Level 3 surface reflectance data. Orthorectification was verified using shipboard
location data, which required small corrections for images two and three.

**2.5 Carbonate chemistry calculations and a synthetic OAE experiment constructed from in-water data**

        All carbonate chemistry calculations presented here were conducted with CO2SYS v3.1.1 run in
the MATLAB environment (Sharp et al., 2023). We used total scale pH ($pH_{tot}$) and the Mehrbach acid dissociation constants refit by Dickson and Millero (option 4 in CO2SYS). Higher resolution underway datasets of $fCO_2$, T, S, and RT, were temporally downsampled to the TA data, taking the closest measurements via timestamp, for a time-matched dataset over the cruise duration. We used this dataset to calculate surface water carbonate chemistry for the entire survey, including $pH_{tot}$ and DIC. We
defined a threshold RT measurement of 0.5 ppb to distinguish between samples taken inside the patch and samples representing the "baseline" outside of the patch, in order to assess the variability of the carbonate chemistry data with respect to measured RT fluorescence. Significant differences between "in patch" and "baseline" data were assessed using a two-way t-test. Relationships between salinity and $fCO_2$, and salinity and TA, were assessed using the "fitlm" model in MATLAB.
We then conducted a "synthetic" OAE experiment in which we used the RT signal to estimate what a similarly scaled alkalinity addition would have looked like, using the MRV framework proposed

above (Fig. 1, eqs. 1a-d). The goal was to use this framework to calculate what the maximum OAE and CDR signal would be in the center of the measured patch over time using in-water measurements. The MATLAB code for this calculation is provided as a supplementary file. We found the maximum RT value within the patch in every hour of the monitoring campaign. The baseline carbonate chemistry values at these points were assessed in two ways. The measured $fCO_2$ -TA pairs inside the patch at these time points defined the "true" baseline. As an alternate approach, we constructed a "dynamic baseline" by linear interpolation between the two nearest out-of-patch points in time. The difference between these "true" and "dynamic" baselines are discussed below.

We assumed the release of 20 metric tonnes (1 tonne = 1000 kg) of NaOH along with the 56 kg of RT, resulting in a TA:RT ratio of 8.9 µmol kg$^{-1}$ per ppb of RT measured. We then added this TA enhancement on to the baseline TA:

$$TA_t = TA_{bl,t} + \Delta TA_{OAE,t} = TA_{bl,t} + 8.9 * RT_t. \qquad \text{(eq. 4)}$$

The subscript "bl" refers to the baseline data measured by the underway system, and "OAE" refers to the addition of alkalinity via the OAE intervention. We then assumed that in a dilution-only scenario, DIC at the beginning of OAE remained unchanged, and we calculated the entire carbonate system using $TA_t$ and $DIC_{bl,t}$. For DIC ingrowth due to gas exchange, we started with only the OAE perturbation and its effect on $fCO_2$ and then calculate the uptake of $CO_2$ and its effect on DIC. Using TA and $DIC_{bl}$, we calculated DIC uptake driven by OAE enhancement as the difference between the flux of $CO_2$ due to natural (baseline) processes, and the flux due to the OAE process:

$$\frac{dDIC_{CDR,t}}{dt} = F_{CO2,OAE} - F_{CO2,bl}$$
$$= \frac{k}{z}K_0(fCO_{2,\ t} - fCO_{2\ atm,t}) - \frac{k}{z}K_0(fCO_{2\ bl,t} - fCO_{2\ atm,t})$$
$$= \frac{k}{z}K_0\left(fCO_{2\ t} - fCO_{2\ bl,t}\right); \qquad \text{(eq. 5)}$$

where the subscript "CDR" refers to the change in the carbonate system due to $CO_2$ uptake from the atmosphere, and $fCO_2$ subscripts are consistent with their definition in Eq. 1c. Here, the difference between the natural and OAE-driven $CO_2$ fluxes collapses to be proportional to the difference between $fCO_2\ _t$ and the contemporaneous baseline seawater $fCO_2$ ($fCO_2\ _{bl,t}$). We used hourly meteorological data from the Buzzard's Bay meteorological tower (BUZM3) over the deployment time period to calculate gas transfer velocity. Windspeed at 10 meter height was calculated from the measured windspeed (Yu et al., 2020), and 10-meter windspeed ($U_{10}$) was then used to calculate gas transfer velocity in cm hr$^{-1}$ (Wanninkhof, 2014). We used the depth of RT penetration from CTD casts as z = 11.4m, and the solubility of $CO_2$, $K_0$, was extracted from CO2SYS using *in situ* underway T,S conditions. We then calculated $CO_2$ uptake every hour, resulting in a small increase in the DIC reservoir that accumulates in the patch of water over time:

$$\Delta DIC_{CDR,t} = \sum_{t0}^{ti} dDIC_{CDR,t} \cdot \text{(eq. 6)}$$

To account for this DIC accumulation on net $CO_2$ uptake, we calculated an updated $fCO_2$ at every timestep t:

$$DIC_t = DIC_{bl,t} + \Delta DIC_{CDR,t} \text{ (eq. 7)}$$

We then recalculated the carbonate system using $DIC_t$ and $TA_t$, resulting in a new $fCO_{2\ t}$ for each timestep that reflected the combination of gas exchange and dilution (see MATLAB script attached for full calculation).

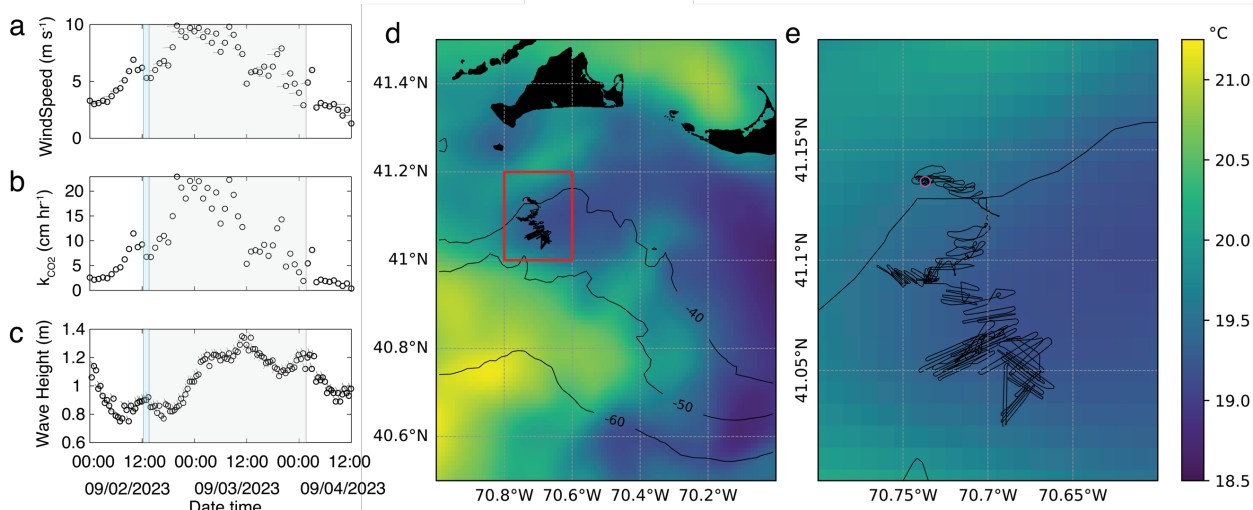

**Figure 2:** Meteorological and oceanic conditions of the study. **a)** shows windspeed measured at the BUZM3 station in Buzzard's Bay, with pointers indicating wind direction. Time is UTC. **b)** shows calculated CO2 transfer velocity from windspeed data. **c)** shows wave height measured at the Buoy 44097 (Block Island), with pointers indicating the mean wave direction. The blue shaded area indicates the dispersal period, and the gray shaded area shows the monitoring period. **d)** shows satellite-based SST in the region from September 3, with contours showing bathymetry in meters. The ship track for the entire monitoring track is shown south of Martha's Vineyard in black. **e)** Shows the red box inset in panel **d** showing the ship track in black, and the exact release location as a pink dot. The color scale for **d** and **e** are the same.

## 3    Results

### 3.1 Baseline oceanic conditions

Weather conditions in late August and early September 2023 along the New England coast were characterized by a series of storms passing through the area, including Hurricane Franklin (initiated August 20), Hurricane Idalia (initiated August 27), and Hurricane Lee (initiated September 5). Wave heights upon vessel departure were relatively low (0.8 m, Fig. 2c) and wind speeds were relatively low at ~3 m s$^{-1}$ (Fig. 2a, BUZM3, NOAA National Data Buoy Center). Over the next 36 hours, wave heights

increased to 1.2 m by the morning of September 3$^{rd}$. Wind speeds increased overnight on the 2$^{nd}$ to 10 m

s$^{-1}$, before falling back down to ~5-6 m s$^{-1}$ by late morning on the 3$^{rd}$. These high and variable windspeeds translated to a mean calculated $CO_2$ gas transfer velocity $k=12\pm6$ cm hr$^{-1}$ over the monitoring period (Fig. 2b), slightly higher and more variable than the interannual mean $k$ for the south of Martha's Vineyard of $9.9\pm1.2$ cm hr$^{-1}$ in summertime (Guo et al., 2025).

Sea surface temperatures varied spatially by about 2 °C in the study area, characterized by several water masses actively moving in the region (Fig. 2d,e). The SST field moved slightly relative to our survey over the 36-hour period, but the overall trend of cooler water to the north and west, and warmer water to the south and east, held for the entire expedition. Pre-site CTD surveys demonstrated a relatively stratified water column with warmer, fresher water at the surface (T~17.5 °C, S~31.7,

$\rho$=1.0225 kg m$^{-3}$) and a mixed layer depth of about 10 meters (Fig. S3).  The carbonate chemistry of the study area, characterized by both underway and CTD measurements, was weakly buffered compared to open-ocean conditions, with a surface pH of ~8.0, an $fCO_2$ of ~450 µatm, and a relatively low alkalinity of ~2143 µmol kg$^{-1}$ (Fig. S3a,b). These conditions are typical for the shelf region in the summertime (Wang et al., 2013; Cai et al., 2020; Hunt et al., 2021). Surface waters contained 10 µmol kg$^{-1}$ POC and

about 0.2 µmol kg$^{-1}$ PIC, for a PIC:POC of ~0.02 (Fig. S3e,f). Phosphate and Nitrate+Nitrite were typically below detection at the surface (<0.015 and <0.04 µmol kg$^{-1}$, respectively), with low silicate of 0.3 µmol kg$^{-1}$ (Fig. S3 d,e).

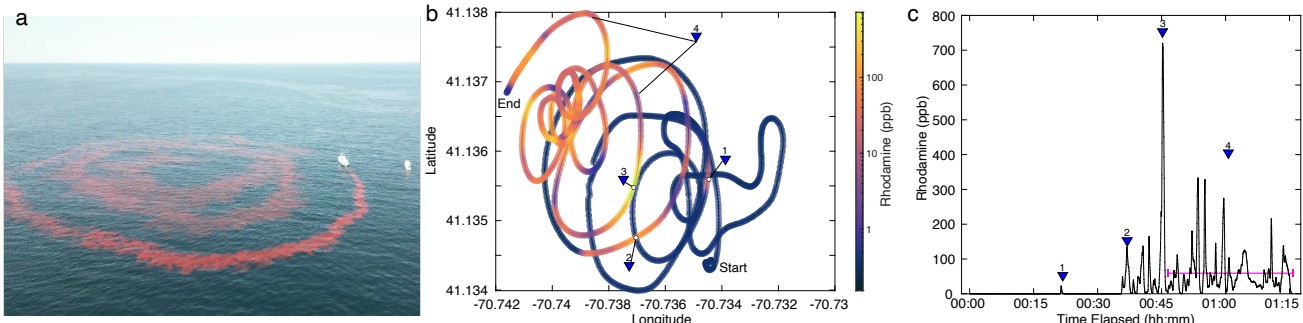

**Figure 3:** Dispersal of rhodamine dye. **a**) A drone image during the dispersal from the *R/V Connecticut*. A second chase boat can be seen adjacent to the *R/V Connecticut*, from which the drone was being operated. **b**) The ship track during the 75-minute dispersal in cartesian latitude-longitude coordinates, colored by underway RT concentration. **c**) The timeseries of RT concentration during the dispersal.  The four triangles indicate RT signals for comparison to the ship wake dilution model of Chou (1996). Triangle 4 refers to the mean RT concentration for the remainder of the release (58 ppb, shown by the pink line).

**3.2 Dispersal**

    Dispersal of the 1,000 liter tote filled with RT solution took approximately 75 minutes to

complete (Fig. 3) resulting in an average RT dispersal rate of 0.2 liters per second. While the gravity feed worked well, the flow rate decreased slowly as the tank emptied. Additionally, the flexible nature of the dispersal hose made the exact release location relative to the vessel difficult to control, and flow variations were common. The spiral pattern was straightforward to follow, given the high visibility of RT in seawater (Fig. 3a). However, surface currents were vigorous enough to displace the patch in

space (Fig. 3b). About 30 minutes through the dispersal, the ship's course was adjusted to disperse on

top of existing rhodamine dye. This helped limit the overall size of the dispersal patch, and also made it possible to measure fluorescence within the patch during the dispersal period (Fig. 3b,c).

We selected four representative dye signals from the dispersal period to compare to the ship wake dilution model (Eqs. 2,3; triangles 1-4, Figs. 3b,c, 4). A fluorescence spike was first detected 25 minutes into the dispersal, after passing over the initial first inner spiral arm (22 ppb, triangle 1, Figs. 3b,c, 4). The second dye signal occurred 35 minutes after the dispersal began, when the ship crossed back over a portion of the dye track that had been dispersed approximately 9 minutes earlier, with a peak concentration of 135 ppb (triangle 2, Figs. 3b,c, 4). The largest spike in concentration, 720 ppb, occurred when the ship returned over a section of the dye track laid in about 22 minutes earlier (triangle 3, Fig. 3b,c, 4). Finally, 26 minutes after the inner rings of dye were dispersed, we started to consistently transit across the patch, sampling elevated RT concentrations, with a mean signal of 58 ppb (triangle 4, Fig. 3b,c, 4). We compared these four concentrations as a function of time since dispersal with established ship wake dilution models from the literature (Chou, 1995, Fig. 4). The mean vessel speed over ground ranged between 1-4 knots, with a mean of 3.8 knots during dispersal (solid red and purple lines, Fig. 4).

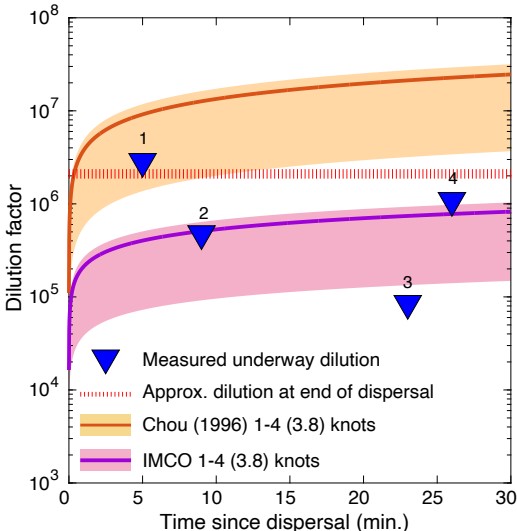

**Figure 4:** Comparison of the ship wake models of Chou (1995) and the IMCO to the measured dilution rates during our dispersal. The range of ship speeds (1-4 knots) is shown in the shaded yellow and pink regions, respectively, with the mean ship speed (3.8 knots) during the dispersal indicated by the red and purple solid lines, respectively. Dilution measured at four time points (Fig. 2c) is shown in the blue triangles. The red dashed line is the dilution calculated for the mass of dye spread evenly across a circular patch with a diameter of 500 meters and a depth of 10m.

Our dilution data fall within and below the two dilution curves, with point 1 accurately captured by the Chou (1996) model, and points 2 and 4 accurately captured by the IMCO model (Fig. 4). The final dilution measurement sits just below the estimated dilution of the entire patch (triangle 4 and red dashed line, Fig. 4), assuming that the 56 kg of dye was dispersed evenly into a patch roughly 500 meters in diameter and a mixed layer depth of 10 m, as estimated from the dispersal imagery (Fig. 3a). Point 3 sits below the IMCO dilution curve, likely reflecting the fact that this spike may be related to sampling multiple "legs" of the dispersal laid on top of each other (Fig. 3b,c).

### 3.3 Monitoring

After the dispersal, monitoring continued for 36 hours. Overall, the patch moved south-southeast by about 8 nautical miles (Figs. 2, 5). The arc-like cyclic pattern of the motion reflects tidal flows. Nine total CTD stations were carried out prior to the dispersal and over the monitoring period. Initially, we planned on pairing "in patch" and "out of patch" CTD casts throughout the monitoring

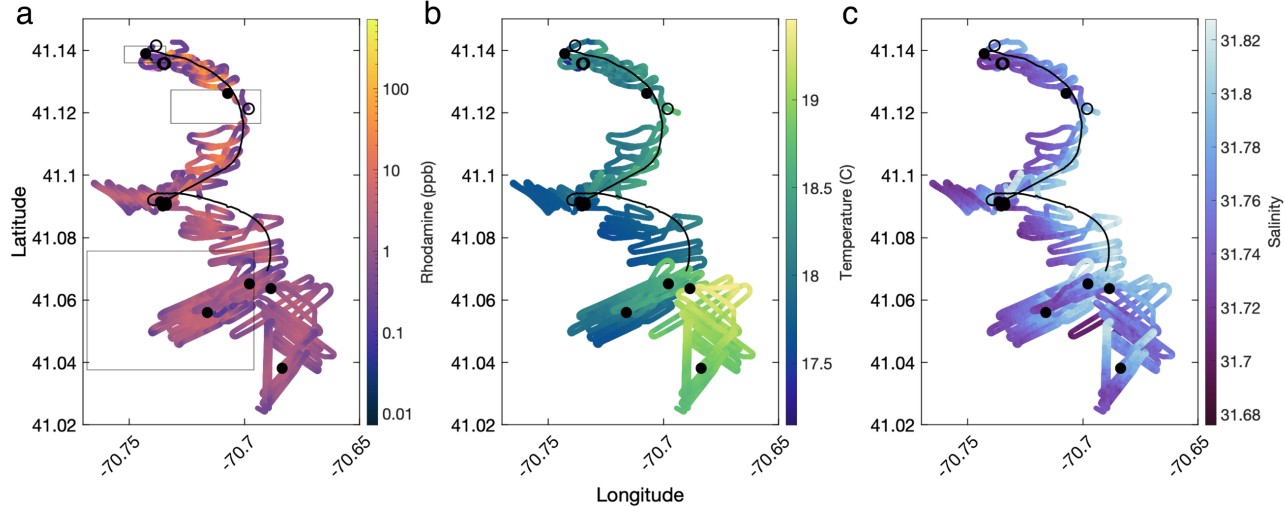

**Figure 5:** Ship tracks displaying underway data from the experiment. In all cases, the black lines show all four drifter trajectories. Filled black circles indicate CTD casts taken within the RT patch. Empty black circles indicate CTD casts taken outside of the RT patch. Panels show the ship track colored by RT concentration (**a**), temperature (**b**), and salinity (**c**). The rectangles in panel (**a**) show the satellite imaging windows presented in Fig. 6.

period (closed and open symbols, Fig. 5). However, the time required for CTD casts, combined with rapid currents in the area, increased the risk of losing the patch during "out of patch" casts, so these
were abandoned after Station 3. Due to the rapid tidal flow, after Station 4 we switched from taking CTD bottle samples to conducting casts without triggering Niskin bottles to collect vertical sensor profiles. The drifter trajectories followed the patch well, both in terms of tidal flow and the mean current direction (black traces, Fig. 5). Because the drifters stayed with the patch, their attached strobes proved valuable for plume tracking overnight when visual identification of dye in the water was
difficult. The real-time RT data readout on the bridge was essential to track the plume once the RT was no longer visually identifiable. Late in the morning on September 3rd, we recovered all four drifters, as increased wave motion was making it difficult to locate and recover these assets. Loss of RT visual signal occurred by the late morning/early afternoon of September 3rd, about 26 hours after the dispersal.

       Surface RT concentration decreased continuously over the ship track, originating in the north,
and traveling south with the mean flow (Fig. 5a), also evident in the satellite and underway data (Figs. 6,7). Surface temperature showed a general trend of warmer water in the beginning of the monitoring period to the north, followed by a cooling in the middle of the ship track, and finally encountering a significantly warmer water mass in the south, towards the end of the survey (Fig. 5b). This general temperature trend is consistent with SST observations for the region (Fig. 2d). Several hours after the
dispersal, we encountered a small but distinct salinity front, with higher salinity to the east (>31.8) and lower salinity to the west (<31.75, Fig. 5c). The patch itself appeared to follow this feature to the south, despite the tidal motion pulling the feature east and west.

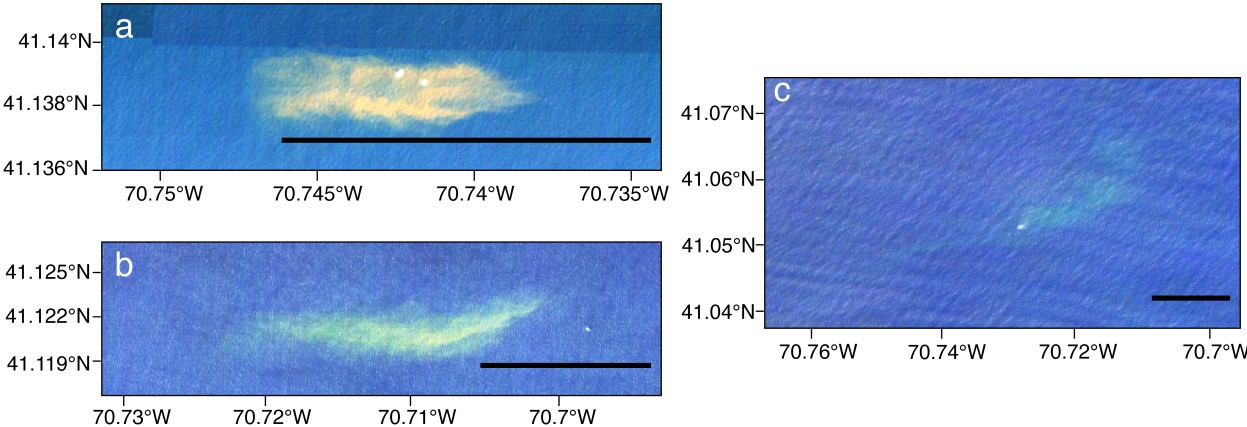

**Figure 6**: Satellite imagery collected via Planet Labs at 3 time points: a) approximately 1 hour after dispersal; b) approximately 6 hours after dispersal; and c) approximately 24 hours after dispersal. Images shown are true color, scaled to enhance image brightness. Scale bars in each image are 1 km.

The patch diluted and dispersed over the monitoring period, but satellite imagery confirms that the plume retained a coherent structure with a distinguishable core, rather than the dye dispersing evenly throughout the region (Fig. 6). The first image, collected one hour after dispersal, demonstrates a rhodamine patch elongated in the east-west direction, roughly 800 meters long and 200 meters wide for a total area of 0.16 square kilometers, equivalent to a circle with a radius of 225 meters (Fig. 6a). Over the first 6 hours, the patch stayed coherent, but distorted into a crescent shape, stretching in the east-west direction and the center bowing to the south (Fig. 6b). The patch core remained close to the center, but slightly west of the centroid of the patch. After 24 hours post-dispersal, the patch was still visible via satellite, with more distortion trending southwest-northeast. There was a long tail to the southwest of the main core of the patch (Fig. 6c). Comparing Figs. 5 and 6, the ship track closely followed the orientation of the patch, suggesting that the ship sampling approach using visual and sensor-based detection, was sufficient to capture the patch distribution over at least the first 24 hours of monitoring.

Underway data plotted as a timeseries shows similar results (Fig. 7). Underway RT concentration decreased over time, with the oscillating values reflecting repeated transits through the patch (Fig. 7a, black dots, high values) into surrounding "baseline" seawater (low values) and back again. The constancy of the highest RT values, and their steady decay over time, suggest that we were sampling the highest concentrations of the patch throughout the monitoring period. Small variations in the minimum RT fluorescence are evident over the monitoring period, and likely arose due to two factors. First, there may have been small changes in the fluorescence of background seawater due to differences in chlorophyll *a* and other fluorescent organic compounds in seawater. Also, as the survey progressed, the ship maintained a closer survey pattern relative to the peak in fluorescence to ensure that we did not lose the patch overnight. Thus, the measured "baseline" during this period was likely still sampling the edges of the patch, rather than a true out-of-patch baseline. Minima in the RT signal can be seen at around or less than 0.1 ppb, suggesting that the effective detection limit was around this value

(Hixson and Ward, 2022). In this application, we define a threshold "baseline" value of <0.5 ppb, which represents a greater than 1,000 fold dilution from the highest initial RT fluorescence of ~720 ppb measured during the dispersal (Fig. 3b,c), and more than a 100 fold dilution from the mean patch fluorescence of 58 ppb at the end of the dispersal (Fig. 3c).


Rhodamine fluorometers deployed on two of the drifters closely matched the highest underway rhodamine concentrations (yellow dots, Fig. 7a), indicating the utility of drifting assets to track and monitor water mass features over 24-48 hours. However, we note three exceptions. At approximately 4 hours, the two drifter RT signals deviated from the peak RT concentration, dropping to ~10 ppb and became highly variable, while the ship continued to transit through high-fluorescence sections of the patch.


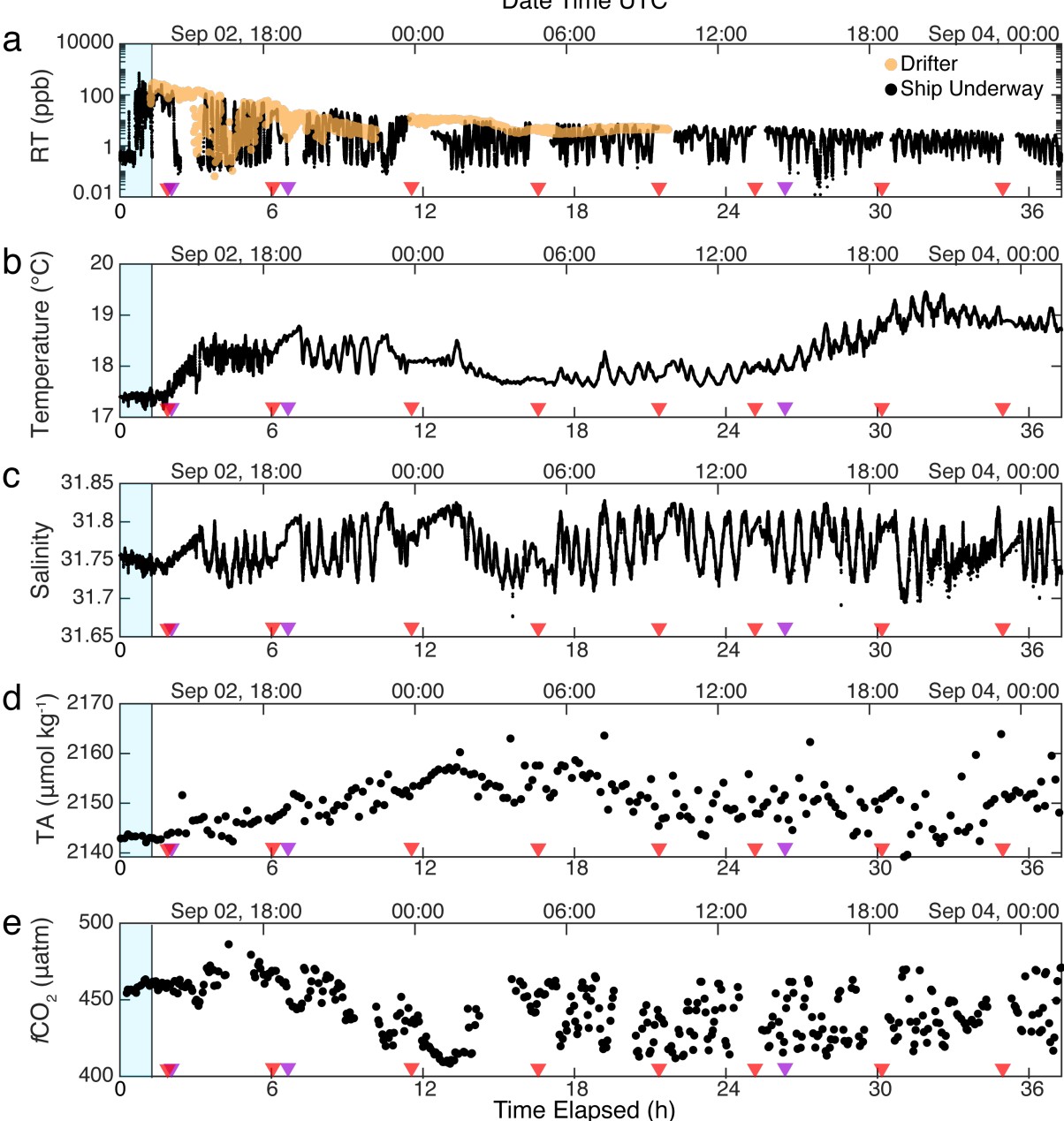

**Figure 7:** Surface data presented as a timeseries, in hours elapsed from initiating the dispersal. Red triangles indicate times when CTD casts were conducted. Purple triangles indicate times of satellite image collection. The blue shaded region indicates the dispersal window; all data collected after this window was during the monitoring period. **a**) shows RT data from the ship underway system (black points) and the lagrangian drifters (yellow points). Temperature (**b**) and salinity (**c**) are shown from the ship's thermosalinograph. Total alkalinity (**d**) and $f$CO$_2$ (**e**) were collected at lower temporal resolution.

The drifter deviation from the patch was also evident in the ship track data between 41.12 - 41.1°N (Fig. 5a) at the peak of a tidal cycle, which shows the drifter traces (black lines) pulling to the

east of the highest RT concentrations measured by the ship. Surprisingly, the drifters began sampling higher RT signals again at just over 5 hours time elapsed, demonstrated by the re-alignment of the drifter tracks with the ship track (Fig. 5a) and the drifter RT signals matching peak RT concentrations once again (Fig. 7a). We interpret this deviation and subsequent re-convergence as a tidal feature and indicates the fundamental importance of tides to small-scale shear and fluid flow in this region.

After about 8 hours, the two drifters began to significantly diverge from the main patch, demonstrated by the decline of drifter RT readings relative to the higher concentrations from the ship's underway system (Fig. 7a). The following gap in drifter data at 10 hours post-dispersal is due to the recovery and re-deployment of the drifters in the afternoon of the 2nd, followed by the final recovery of the drifters on September 3rd. After the drifters were re-deployed in the center of the patch at ~11 hours,

the vessel began to sample a lower-concentration arm of the patch, while the drifters appeared to track the center of the patch (Fig. 5a). After about 13 hours post-deployment, the drifters consistently stayed near the upper end of the measured underway RT fluorescence, suggesting that both the ship and the drifters were sampling the center of the patch.

Variability in temperature and a salinity front were observed clearly in the timeseries data,

independent of our traversing in and out of the RT patch (Fig 7 b,c). Analogous temporal and spatial variability was visible in underway TA (Fig. 7d) and $fCO_2$ (Fig. 7e). Total alkalinity started at about 2143 µmol kg$^{-1}$ during the dispersal and climbed to about 2157 µmol kg$^{-1}$ about 12 hours after the dispersal, during which time the signal became significantly more variable. Similarly, a relatively stable $fCO_2$ became significantly lower and more variable over time (Fig. 7e). The variability at this stage may

be associated with the higher wind speeds and swells that we encountered starting in the later part of September 2nd, and continuing into September 3rd, and/or due to the salinity front. After about 36 hours, we concluded the monitoring phase and transited back to port. Upon departure, the patch was still clearly measurable with a signal of 4-5 ppb above a measured baseline of ~0.1 ppb at that period of the survey (Fig. 7a).

Over the cruise, the mixed layer depth defined (defined as a difference in potential density from the surface of 0.03kg m$^{-3}$, Jones et al., 2014) was 5.5±2.9 m. The vertical distribution of RT, captured by vertical CTD profiles, was consistently deeper than the mld with a penetration depth of 11.4±2.4 meters over the duration of the campaign, following the S=31.8 contour (Fig. 8). High concentrations following the release diluted over time, with a higher concentration ~10 ppb being retained at the

surface until 4 AM UTC on September 3rd. Near the end of the section, we observed a shoaling of the rhodamine signal and further dilution, which coincided with the appearance of the warmer 19°C surface water mass at about 30 elapsed hours (Fig. 5b, 7b). The chlorophyll maximum, defined by the strongest Chl $a$ fluorescence, was found below the mixed layer at 15-25 meters deep, likely because strong stratification limited nutrient supply to the sea surface (Cornec et a., 2021, Fig. S4). Because of the

spectral overlap in Chl $a$ and RT fluorescence, we used baseline CTD casts to construct a Chl-$a$ correction to the RT fluorescence data (Fig. S5). This small correction (0.24 ppb RT mg$^{-1}$m$^3$ Chl $a$) does not significantly change the distribution of RT because Chl-a was largely absent from the sea surface. The lack of Chl $a$ in the seasonally stratified mixed layer in late summer further suggests that OAE deployments during this season would limit the interference with phytoplankton communities residing

below the mixed layer, as they would be separated vertically from the highest alkalinity signals at the sea surface.

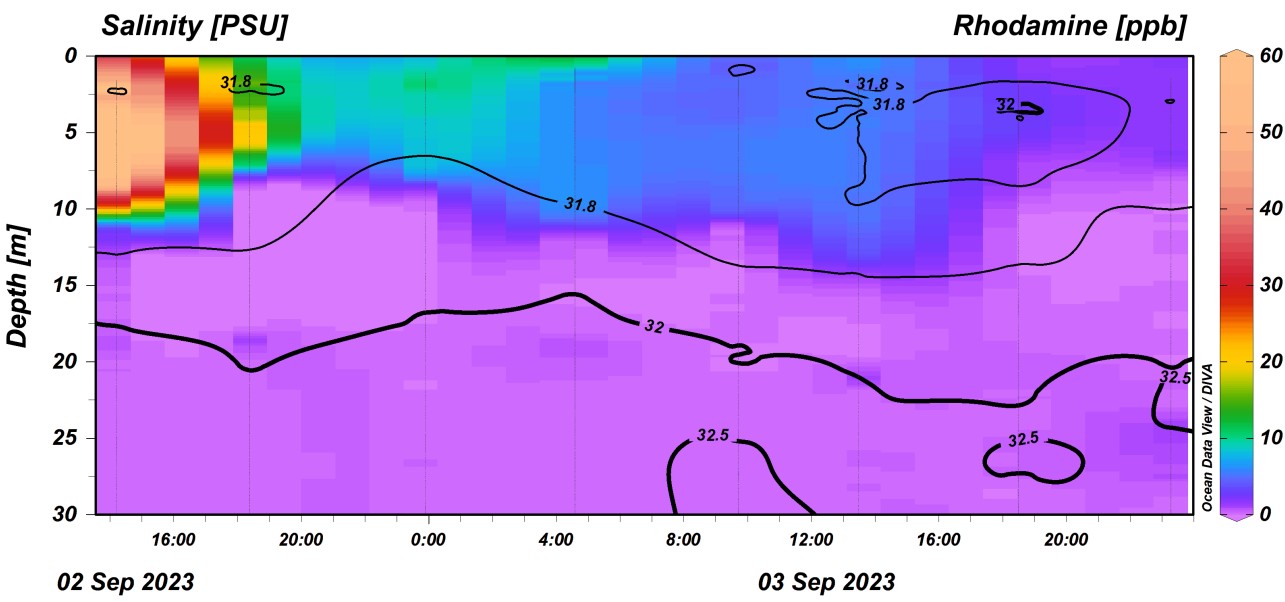

**Figure 8:** ODV Section of chlorophyll *a*-corrected rhodamine concentration collected from the CTD as a function of time (UTC). Black contours show salinity values.

## 3.4 Background carbonate chemistry analysis


After downsampling of higher-resolution samples (T, S, *f*CO₂, and RT) to the TA data, the dataset contains a total of 214 paired measurements of T, S, *f*CO₂, TA, and RT (Table 1). Out of these samples, 168 (78%) were taken within the patch and 46 samples (22%) were collected in "baseline" conditions. Critically, our monitoring approach sampled similar values of all physical and chemical parameters in terms of the mean and variance, both within the patch, and outside of the patch (Fig. 8, Table 1). This was the expected pattern since no alkalinity was added during the dye dispersal. The "baseline" *f*CO₂ samples are slightly higher than the entire dataset, and than the "in-patch" samples,


**Table 1:** Underway data means with one standard deviation for samples collected from outside of the patch (Baseline) and inside the patch. The differences between in-patch and baseline *f*CO₂ (p=0.4) and TA (p=0.15) are not significant.

| Location | *f*CO₂ (µatm) | TA (µmol kg⁻¹) | T (°C) | S | RT (ppb) |
|---|---|---|---|---|---|
| Baseline (n=46) | 454±22 | 2148±6 | 18.0±0.6 | 31.76±0.05 | 0.3±0.1 |
| In-patch (n=168) | 446±18 | 2150±8 | 18.2±0.6 | 31.75±0.06 | 8.4±21.6 |
| All (n=214) | 448±19 | 2149±8 | 18.2±0.6 | 31.75±0.06 | 6.2±18.8 |

largely driven by the four high values measured at approximately 5-6 hours after the dispersal (Fig. 7e, 9b,c). However, the mean and variance in $fCO_2$ for all data categories is similar, and are statistically indistinguishable (p=0.4, Table 1). Some of the variability observed in carbonate chemistry data can be explained by a correlation with salinity (Fig. 9). Alkalinity is known to covary with salinity in a semi-conservative manner in this region (Wang et al., 2017; McGarry et al., 2021, Hunt et al., 2021). The calculated TA-S slope for our dataset was $69\pm12$ µmol kg$^{-1}$ (S unit)$^{-1}$, with an $r^2 = 0.15$ (Fig. 9a, S6a).

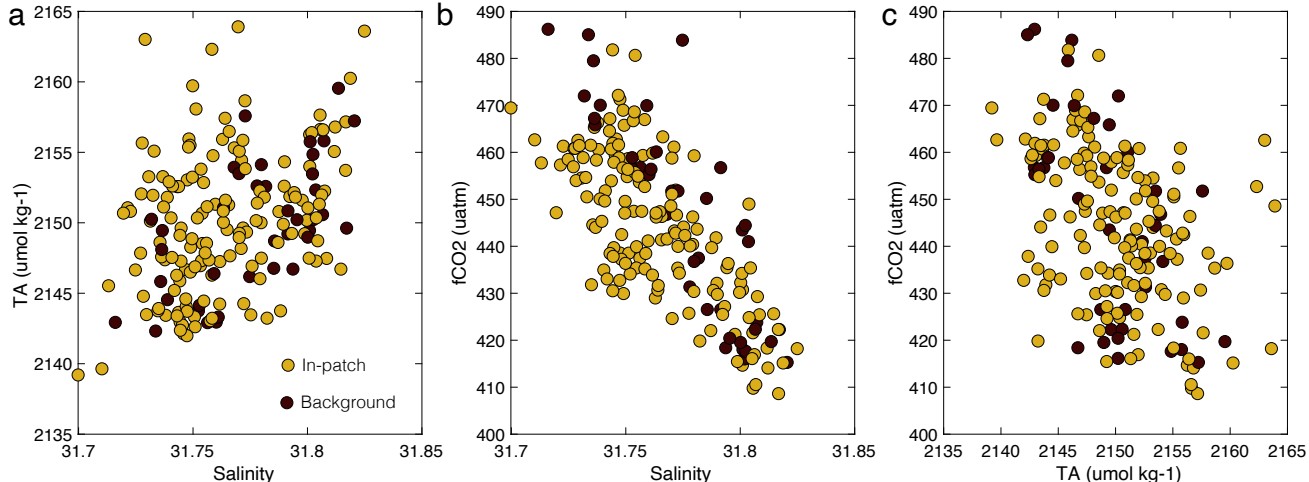

**Figure 9:** Carbonate chemistry data split out by in-patch (RT> 0.5 ppb) and out-of-patch, or background data (RT<0.5 ppb). Cross-plots are constructed for TA versus salinity (**a**), $fCO_2$ versus salinity (**b**), and $fCO_2$ versus TA (**c**).

This slope is at the upper end, but within error, of historical surface water TA-S relationships for the Gulf of Maine and Southern New England/Georges Bank regions analyzed over much larger salinity ranges (18.4-63.3 µmol kg$^{-1}$ (S unit)$^{-1}$; Hunt et al., 2021). The $fCO_2$-S relationship demonstrated a slope of -466 µatm (S unit)$^{-1}$ and an $r^2= 0.54$ (Fig. 9b, S6b). For comparison, the $fCO_2$-T relationship was poorly constrained ($r^2=0.03$) with a slope of -6.2 µatm °C$^{-1}$ (p>0.01; not shown). As expected, $fCO_2$ and TA were inversely correlated with each other (Fig. 9c), with higher TA corresponding to lower $fCO_2$.

## 4 Discussion

### 4.1 Dispersal and dilution

Because we doubled back over the RT patch during dispersal, the ship's underway data could be used to interrogate the near-field dispersal behavior of material in the wake of the R/V Connecticut (Fig. 4). Ship-wake dilution models explicitly separate near-field from further dilution regimes (Chou, 1996), and focus on empirical dilution in the near-field (e.g. the ratio between measured and initial concentration; eq. 2) rather than attributing changes in concentration to specific processes such as eddy diffusivity or advective mixing. Given the turbulent nature of near-field mixing, we anticipate that almost all of the mixing was driven by the ship wake, and not by horizontal diffusivity (with a lateral eddy diffusivity of ~5 m$^2$ s$^{-1}$, Rypina et al., 2019); vertical diffusivity (~10$^{-4}$ m$^2$ s$^{-1}$, Rypina et al., 2019) or molecular diffusion (~10$^{-9}$ m$^2$ s$^{-1}$, Zeebe, 2011).

The general agreement between data and model is surprising given the scale difference between the large container ships for which these models were developed, and the much smaller *R/V Connecticut*. We further note that the dispersal pipe was dragged about 20 feet behind the vessel and at 700 the surface, far from the high-velocity zone directly behind the ship's propeller. Thus, we should expect measured dilution to be consistently less than the model predictions, and indeed many of the dilution values fall below the dilution model of Chou (1996). This model also assumes that discharge occurs behind a vessel moving in a straight line, rather than in a spiral pattern with overlapping layers of dispersed material. Nonetheless, this model predicts a similar order of magnitude of dilution to our 705 measurements, suggesting that existing ship wake models, in a mean sense, can help to guide dilution and dispersal behavior for liquid alkalinity addition to the surface ocean from vessels.

One important caveat is that there is significant patchiness in measured concentration while transiting over the dye patch, with one very high peak (720 ppb) appearing midway through the dispersal (Fig. 2c). This spatial variability in concentration highlights the highly heterogeneous 710 dispersal field in the turbulent ship's wake, and suggests that higher resolution models of ship-wake dilution are required to fully understand the turbulence field at these small scales. In addition, it also suggests that dilution via ship wake alone may not be enough to fully mix away small patches of elevated alkalinity water over the timescale of minutes. Such large excursions will likely have a more extreme impact on any organisms living in this water, and would also potentially result in the 715 precipitation of calcium carbonate and brucite that would remove alkalinity from seawater, affecting the overall efficiency of OAE for CDR (Moras et al., 2022, He and Tyka, 2022, Hartmann et al., 2022).

Current dilution models do not capture any biogeochemical impacts beyond dilution, although they could be amended to include reaction rate terms (Chou, 1995). Specifically, for OAE, the mineral precipitation thresholds discussed above may be exceeded when dispersing high-pH solutions into 720 seawater, and these must be considered against dilution timescale to assess the efficient transfer of alkalinity from the ship into the dissolved phase (He and Tyka, 2022). Furthermore, ship wake-induced turbulence may impact air-sea interactions. For example, ship wakes produce both high amounts of turbulence and bubbles (Nylund et al. 2021), both of which can enhance gas exchange and may potentially increase the amount of $CO_2$ being taken up in the ship's wake. The interaction between ship 725 wake turbulence, bubble production, and air-sea gas exchange, as well as its effect on mineral precipitation and biological activity, should be further investigated with respect to ship based OAE applications.

**4.2 Defining a suitable baseline for in-water CDR calculations**

In-water MRV frameworks require careful consideration of baselines, which must be established either from historical data, or from in-water data sampled at the same temporal and spatial resolution as the intervention itself. It is against this baseline that additionality is assessed, both for OAE and for subsequent CDR (Fig. 1). One critical aspect of assessing baselines is understanding their variability. While it has been argued that large variability in carbonate chemistry places limitations on the viability 735 of in-water MRV of OAE (Ho et al., 2023), there are defined oceanographic drivers and structures to

this variability that may help to reduce the effect of variability on CDR quantification from in-water data.

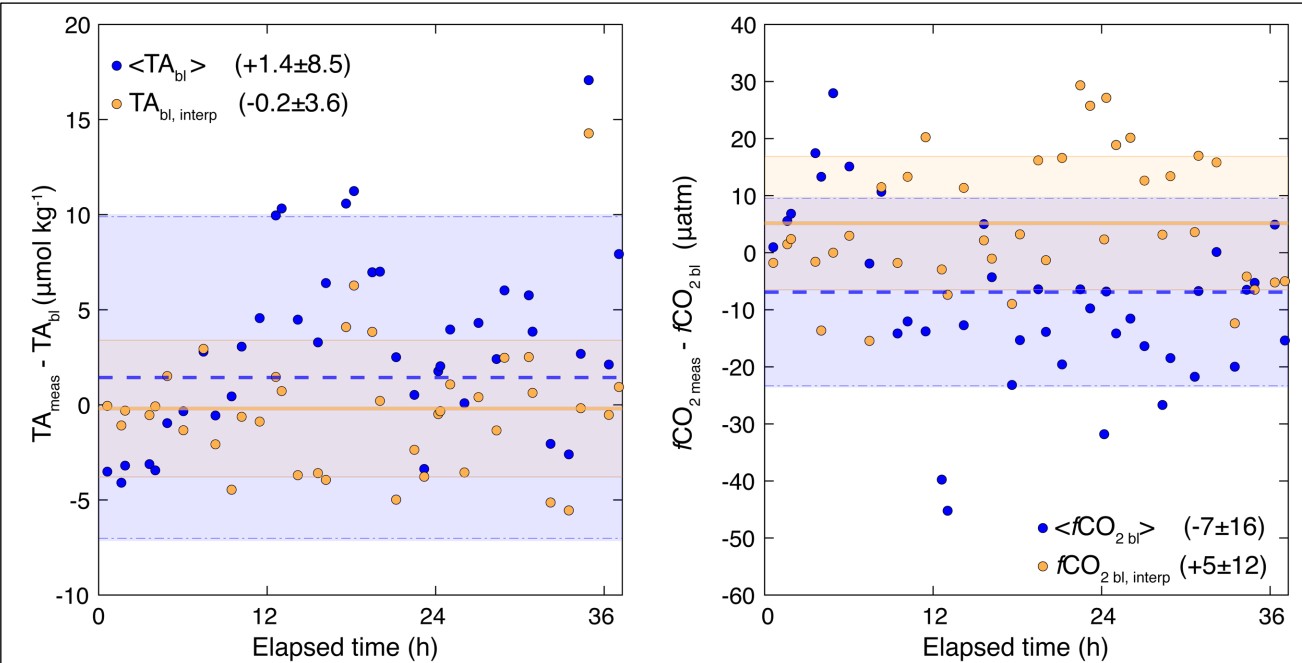

**Figure 10**: Analysis of baseline definitions for hourly TA (a) and $f\mathrm{CO_2}$ (b) used in the synthetic OAE experiment. In each panel, the difference between measured in-patch values and the mean out-of-batch baseline (e.g. $<\mathrm{TA_{bl}}>$, Table 1) is shown in blue circles, with the mean and standard deviation of the offset is shown by the blue dashed line and blue-shaded box. The difference between the measured in-patch values and a dynamic interpolated baseline (e.g. $\mathrm{TA_{bl,\,interp}}$) is shown in yellow circles, with the mean and standard deviation of the offset shown by the solid yellow line and yellow-shaded box. Means and standard deviations of these offsets are also shown in the panel legends.

Establishing a complete baseline scenario for this experiment required interpolation of the higher-resolution data to the lowest-resolution sample (Subhas et al., 2023), which in this case is
underway TA, sampled approximately every 10 minutes. This downsampling thus limits the overall ability to construct a baseline scenario in time, and the ability to sample small spatial scale features. On a vessel transiting at 4 knots, this 10-minute sampling frequency translates to a sample taken every 1200 meters travelled. In contrast, a sampling frequency of 1 Hz translates to a sample taken every 2 meters travelled. Clearly, for small-scale interventions and with limited platform options, higher sample
resolution is preferable. In some cases, data resolution may ultimately limit the ability to observe both the intervention and the baseline. Given these limitations in space and time with currently available sensors, we suggest that in future monitoring campaigns, a concerted effort should be made to consistently sample the true baseline (defined via the water tracer reading in background seawater) as frequently as possible.

Some of the features we observed occurred at very short spatial scales, such as the temperature and salinity fronts that we encountered over the 36-hour monitoring period (Figs. 5,7). The TA-S relationship was largely in line with published regional relationships (Fig 9a, Table 1, and Section 3.4), and normalizing TA to salinity (e.g. dividing the measured TA by S and multiplying by S=35) removed any relationship between TA and S (Fig. S6c). Thus, TA variations over the timescale of this

experiment were likely solely driven by variability in salinity and a conservative regional TA-S relationship. On the other hand, the slope of $f$CO$_2$-S resulted in a gradient of almost 90 µatm across the 0.1-unit salinity front (Fig. 9b). This relationship was not driven by temperature, as demonstrated by the large gradient in temperature-normalized fCO$_2$ as a function of salinity (Fig. S6d), and instead was likely driven by distinct biogeochemical characteristics of these two water masses. Such small-scale

variability is not surprising given sluggish CO$_2$ equilibration timescales and the documented response of $f$CO$_2$ (as opposed to DIC or TA) to biological productivity in coastal U.S. waters (Cai et al., 2020). It is therefore critical to sample $f$CO$_2$ directly during OAE monitoring campaigns, as this property demonstrates significant biologically driven variability, and is the parameter that is directly used in the CDR calculation (eq. 4).

Because we conducted a tracer-only experiment, we sampled the "true" baseline conditions throughout the entire campaign. Using RT as a delineation, we have demonstrated that the variability in carbonate chemistry is similar inside and outside of the tracer patch (Table 1, Fig. 10). In contrast, actual OAE experiments will only be able to sample outside of the patch to establish a baseline against which to assess CDR additionality. We assessed the applicability of two distinct baselining scenarios for

implementation in our OAE analytical framework (Fig. 10). First, we examined the offset between measured in-patch TA and $f$CO$_2$ samples and the mean value measured outside of the patch over the entire 36-hour experiment (Table 1, blue circles, Fig. 10). Second, we examined a "dynamic baseline" approach where we calculated an in-patch baseline as a linear interpolation between the two nearest out-of-patch samples taken over time (yellow circles, Fig. 10). The mean offset approach results in an offset

of -1.4±8.5 umol kg$^{-1}$ for TA (Fig. 10a) and -7±16 µatm for $f$CO$_2$ (Fig. 10b), largely reflecting the variance in the entire measured dataset (Table 1). The dynamic interpolation improves the offsets in size and reduces overall variability, resulting in an offset of -0.2±3.5 umol kg$^{-1}$ for TA (Fig. 10a), and 5±12 µatm for $f$CO$_2$ (Fig. 10b). Thus, this dynamic baseline approach improves the accuracy of the baseline calculation, and reduces the influence of variability by (8.5-3.5)/8.5 = 60% for TA and (16-12)/16 =

25% for $f$CO$_2$. We perform the OAE and gas exchange calculations (section 4.3) using both the true in-patch measurements, and the dynamic interpolated baseline, to compare how much uncertainty the baselining approach contributes to the overall CDR estimate.

**4.3 Synthetic OAE experiment and extraction of a CDR signal from in-water monitoring data**

With an established baseline, we carried out a hypothetical OAE experiment, using a fixed ratio of TA:RT and our measured signals of RT fluorescence over time. The synthetic OAE signal in TA, calculated from measured RT concentrations (eq. 3), gets diluted over time, starting at about 3200 µmol kg$^{-1}$ (an enhancement of about 1,000 µmol kg$^{-1}$, Fig 10a,e) during the dispersal, and dropping over time to a final enhancement at 36 hours of just under 20 µmol kg$^{-1}$ (Fig. 11e). This signal is readily

detectable given the variability and signal to noise in our measurement system (Fig. 8, 10 Table 1). Similarly, $f$CO$_2$ is initially very low, and climbs back to baseline values (Fig. 11b), with a deficit

relative to baseline of 20-40 μatm at 36 hours (Fig. 11f). The pH data also indicate a measurable OAE signal of 0.01 to 0.04 units by the end of the monitoring period. Given these results, we expect that the direct OAE signals should be easily detectable for several days, even with small-sized field trials. These measurable signals are in part due to the selection of a site with shallow summertime mixed layers and high retention at the ocean surface (Guo et al., 2025). In general, the Northeast U.S. Shelf also exhibits lower TA than the open ocean (Cai et al., 2020, Hunt et al., 2021), meaning that these coastal waters are less buffered; more sensitive to OAE; and exhibit shorter $CO_2$ uptake timescales than e.g. locations in the gyres (Jones et al., 2014).

The calculation of a CDR signal in pH and $fCO_2$ cannot be made as a comparison of flux relative to the baseline, as is done with model-based methods, but instead must be relative to the null hypothesis of dilution without any gas exchange, where the OAE patch spreads and dilutes but does not interact with the atmosphere. In other words, the accumulation of a CDR signal occurs via active gas exchange, and is calculated by the difference between the "OAE" and the "OAE+CDR" scenario (e.g. yellow minus the blue values, shown as purple diamonds for $fCO_2$, pH, and DIC, fig. 11f,g,h, respectively). These CDR signals, while barely visible for $fCO_2$ and pH in fig. 11f and 11g, are estimated to be 4 μatm for $fCO_2$ and 0.0037 $pH_{tot}$ units, respectively, with the accumulation of DIC approaching 1.8 μmol kg$^{-1}$, at 36 hours.

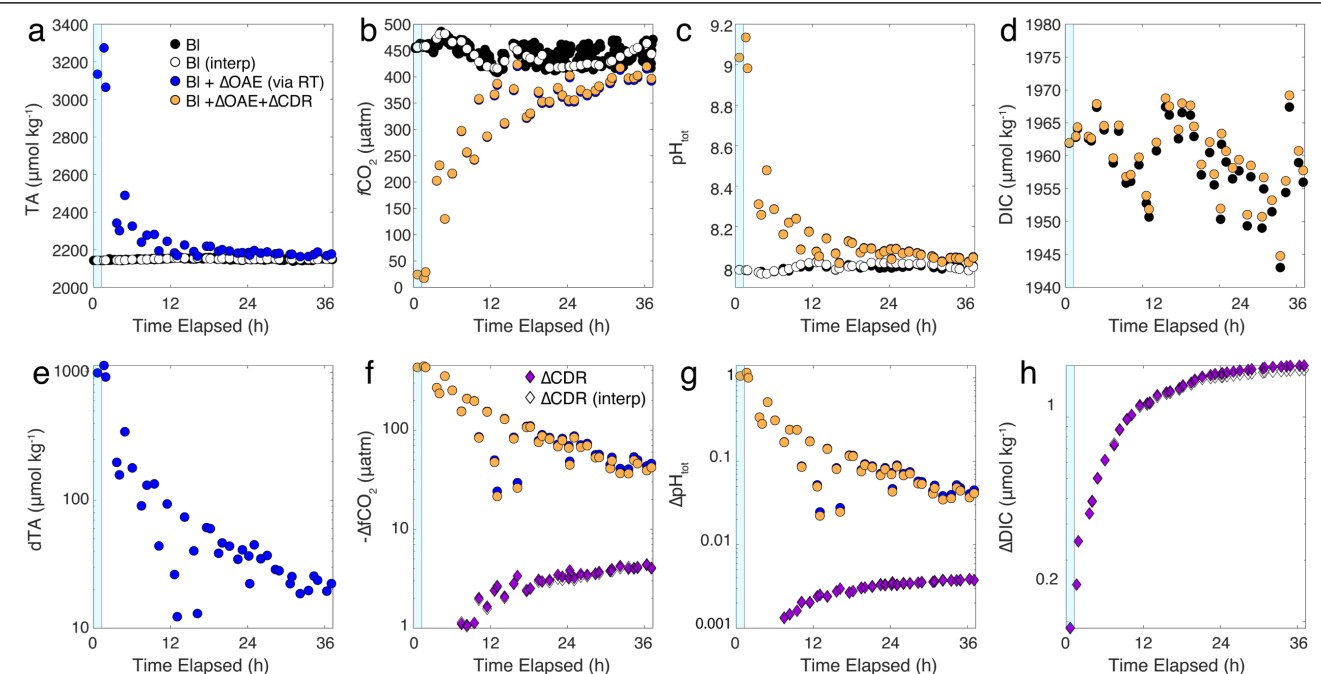

**Figure 10**: A synthetic OAE field trial constructed using the ship's underway data. Baseline data are shown (black points) along with an interpolated "dynamic baseline", calculated from the out-of-patch samples taken (white points, Fig. 7, Table 1). The dispersal period is shown in the blue bar on the left side of each panel. The RT concentration is used to calculate an OAE enhancement signal (blue points). A gas exchange model is used to estimate the CDR capacity of the patch as it takes up $CO_2$ from the atmosphere (yellow points). Measured and modelled values are shown for TA (**a**), $fCO_2$ (**b**), $pH_{tot}$ (**c**), and DIC (**d**). Differences from the baseline, plotted on a logarithmic y-axis, are shown for TA (**e**), $fCO_2$ (shown as its inverse as gradients are negative, **f**), $pH_{tot}$ (**g**), and DIC (**h**). The CDR signal is calculated as the difference between the blue (Bl+ΔOAE) and yellow (Bl+ ΔOAE+ ΔCDR) points, displayed for calculations using the "true" in-patch baseline data (purple diamonds) and the interpolated "dynamic" baseline (open diamonds). Because DIC does not change in the dilution-only case, the difference from baseline for the gas exchange model, and for the difference between the dilution-only case and the gas exchange model, are equivalent.

The calculated parameter values using the interpolated "dynamic baseline" are shown as white circles in Fig. 10a-d 10f-h, with the CDR calculation shown as open diamonds. This calculation falls almost exactly on top of the "true" baseline calculations, with essentially indistinguishable signals in $fCO_2$ and pH (Figs. 11f, g) and a total DIC uptake calculated as 1.7 µmol kg$^{-1}$ at 36 hours (Fig. 11h). The small offset between these two baselining methods, and their slow deviation over time, is likely related to the small and statistically insignificant offset in $fCO_2$ of 5 µatm between the true measured and dynamic baselines (Fig. 10b), resulting in a loss of accuracy in the overall CDR estimate on the order of 0.1 µmol kg$^{-1}$ in DIC over 36 hours, corresponding to a 6% error. We also note that this mean offset only begins to matter when the $fCO_2$ gradients are small enough for this mean offset to become

significant for the overall flux. The short-term (e.g. hourly) variability around this mean offset does not introduce significant errors into the overall CDR calculation because $CO_2$ uptake is a relatively sluggish process with a timescale of days to weeks. Instead, if processes such as dilution and diffusion do not completely remove alkalinity from the ocean surface, and a measurable gradient is maintained, then a CDR signal can be calculated from direct measurements using the proposed dynamic baseline approach. Further refinements in this method will lead to improvements in the estimated CDR through direct measurements.

We can use these model results to estimate the overall efficiency of CDR over the duration of the experiment by comparing the modelled DIC uptake per mole of TA to the theoretical storage capacity of the region (dDIC/dTA = 0.9). Dividing the modeled DIC increase of 1.8 μmol kg$^{-1}$ at 36 hours (Fig. 11h) by the corresponding TA enhancement (19.5 μmol kg$^{-1}$, Fig. 11e) results in a modeled dDIC/dTA of 0.092. Thus, over the 36 hour experiment, we calculate that 0.092/0.9 = 10% of the total potential CDR would have occurred over the first 36 hours in this experiment. Compared to the hypothetical 20 tonnes of NaOH added in this calculation, 10% CDR translates to 2.0 tonnes of $CO_2$ removed from the atmosphere in 36 hours. Although this is a relatively small portion of the total CDR, other models have predicted CDR via OAE to take months or years (Zhou et al., 2024, He and Tyka, 2023). Thus, the potential for directly observing nearly 10% of the total CDR, on the timescale of days, represents a critical opportunity to ground-truth the CDR potential of OAE through in-water campaigns.

We note that this calculation only focuses on the peak RT concentration of the patch, while CDR would be occurring across the entire patch area which is more diffuse and lower concentration (Fig., 6, He and Tyka, 2023). MRV approaches that can accurately capture the entire patch budget will perform better than those that can only sample sparsely across the patch, and critically, early experiments can use a combination of dye tracers and satellite imagery to capture this variability, over time, *in situ* (Fig. 6). Such satellite imagery can serve as a critical link to extrapolate in-water measurements beyond the capabilities of limited in-water assets, and calculate total alkalinity and CDR budgets. This experiment demonstrates that in-water and remote sensing data can be compared with model results, providing an essential validation of the utility of model-data intercomparison for mCDR MRV.

Furthermore, we note the high sensitivity of the deployed RT fluorometers, with the ability to detect signals of several ppb or even lower for extended periods of time (Fig. 5a). The signal to noise ratio of 40-50 suggests that monitoring campaigns should extend for much longer than the one presented here. More critically, as long as the water mass remains at the surface and does not get physically mixed away, these results demonstrate the potential for an overall CDR budget to be calculated directly from in-water measurements. These in-water observations can be further bolstered by remote sensing and aerial imagery. This approach provides an independent assessment of the theoretical effectiveness of OAE for CDR, and illustrates the importance of in-water measurements not just for the validation of regulatory thresholds, but for the direct quantification of $CO_2$ uptake. The framework and approach described here should be validated with real in-water OAE field experiments, for a robust and quantitative comparison of in-water and in-silico CDR estimates.

### 4.4 Comparison with a regional model

Data from this field experiment showed that the dye signal decreased from peaks of ~120 ppb at the end of the release to peaks of ~5-6 ppb after 24 hours (Figs. 3c,5a), corresponding to a decrease in

concentration of 96%. The vertical distribution of Rhodamine dye from the trial indicated a penetration depth of 11.4±2.4 meters (Fig. 8). Overall, the patch traveled south and at the end of the monitoring period, was 12.8 km south/southeast from the original release location (Figs. 2, 5).

To assess whether these findings are representative of broader environmental conditions, we compared them with the Northeast Shelf and Slope (NESS) model, which is based on the Regional Ocean Modeling Systems (ROMS) framework (Chen et al., 2022; Guo et al., 2025). This extensively-validated modeling framework allows us to evaluate how tracer dispersion behaves under different environmental conditions over multiple years, resulting in a modeled tracer-release climatology for this specific study site. Our multi-year ROMS simulations (2009-2017) show that dye concentrations decreased by $97.0 \pm 2.4\%$ within one day post-release in September, aligning with the 96% reduction observed in our field experiment. Similarly, the model simulates that dye vertically dilutes to 0.5% of its initial concentration within $8.0 \pm 2.7$ meters, comparing well with the 11-m RT penetration depth observed in the experiment. The model also simulates a centroid shift of $10.1 \pm 4.3$ km to the east/southeast, again comparable to the 12.8 km shift observed in the field.

The agreement between the observed tracer behavior and the multi-year model simulations suggests that the environmental conditions during the field experiment were consistent with long-term regional circulation. The close agreement between the field experiment and model increases confidence in using both approaches to study tracer dispersion and large-scale transport processes. It also highlights the reliability and suitability of the NESS-ROMS framework for validating tracer release experiments with observational data (Guo et al., 2025). Finally, it highlights that while ocean models can capture mean tracer fields, their dispersal, and overall characteristics, there will be significant year-to-year, seasonal, and even daily differences in ocean state compared with observations. Therefore, in-water conditions may not be sufficiently captured by a single model run. We suggest that model climatologies be used to evaluate the behavior of models with respect to in-water observations, and for assigning uncertainties to CDR calculations.

## 4.5 Practical recommendations for MRV of small-scale OAE experiments and deployments

Based on this experiment, we offer the following recommendations for how to approach in-water MRV for small-scale OAE experiments, to maximize the signal measured over time and the information extracted from those signals to assess CDR. We acknowledge that the approach outlined here is likely not suitable for routine, mature MRV. However, detailed, in-water research is a necessary and critical component of early field projects, and should continue to be a focus of all groups attempting to quantify the CDR potential of OAE during new stages of deployment. We note that the framework proposed here is likely best suited for ship-based or coastal liquid alkalinity deployments, and may need adjustments if solid feedstocks are added to the water column or to sediments.

**A water tracer is critical.** For early deployments, a physical water tracer, such as RT used here, is critical for disentangling mixing effects from chemistry changes, and for assigning water tracing results to water mass features such as salinity, density, mixed layer depth, and temperature. Visible and fluorescent tracers offer the advantage of improved dispersal monitoring and tracking, as they can be easily identified and rapidly sensed. In addition, budgets can be established using remote sensing imagery, either through drones or satellites. This approach is advantageous because, when co-released with alkalinity, an OAE and CDR budget for the experiment can be established through time.

**Lagrangian assets can be used effectively to track the plume.** Drifting buoys successfully tracked the patch during short (~12 hour) deployments, demonstrating their effectiveness for in-water surface measurements of the patch. We suggest that such drifters can be outfitted with sensors and deployed both within and outside of the patch, resulting in a dynamic baseline sampled at identical temporal resolution. The caveat is that currents, tides, and winds may advect and disperse the drifters relative to the patch, which must be assessed and dealt with for each deployment. Furthermore, drifting assets are often considered "discharge" and must be collected after every deployment to ensure that the environmental impact of these deployments is minimal.

**Vertical and horizontal sampling is recommended.** Spatial sampling is important both horizontally, to capture the distribution of patch spreading and the associated concentration gradient, and vertically, to constrain the retention or loss of signal across the mixed layer/pycnocline. As found here, the penetration depth of the tracer was deeper than, but paralleled, the mixed layer depth defined by seawater density. Thus, tracking the vertical extent of alkalinity and tracer is critical to performing budget calculations. Simultaneous measurement of temperature and salinity along with water tracer and carbonate chemistry is recommended. Due to the difference in horizontal and vertical diffusion rates, horizontal sampling should occur at higher resolution than vertical sampling. In the case of solid feedstocks, vertical sampling becomes critical as the interaction between sinking velocity and dissolution rate will produce a profile of alkalinity generation throughout the water column (Feely et al., 2002, Sulpis et al., 2021, Subhas et al., 2022).

**Collect measurements of multiple carbonate system parameters.** We suggest a combination of high-resolution surface water sensors for both the water tracer and at least two carbonate chemistry parameters to maximize information and ensure that all data streams can be interpolated to a common timeframe at reasonable resolution. Further, we suggest that TA sampling be prioritized, given its fundamental relationship to OAE. By combining TA and pH or $fCO_2$ (or all three), both OAE and CDR signals can be diagnosed from underway data. Because of its high variance and its centrality to the CDR calculation, we suggest that measurements of $fCO_2$ over pH are prioritized, until such time that a framework linking pH with CDR is established. We do not recommend combining pH and $fCO_2$ given their covariance with respect to other carbonate system parameters and the resulting high uncertainty in DIC calculations (Dickson et al. 2007, Millero, 2007; Schulz et al. 2023).

**Take samples at the highest resolution possible.** High-resolution sampling offers significant advantages for rapid decision-making during dispersal activities and during subsequent monitoring. While bottle samples from traditional CTD rosettes are often of the highest quality, they also require the most labor and are collected at low resolution. Ten-minute sampling appears to be sufficient to diagnose OAE signals in open ocean conditions, which is possible with current technology. Both pH and $fCO_2$ can be measured at higher resolution (1 minute or faster), with T, S, and RT capable of sampling at much higher resolution (1 Hz or better). Real-time data readouts are highly recommended for making decisions and plume tracking. Given the available sensors, $fCO_2$ or pH would be the most suited for real-time tracking of the carbonate system during an OAE deployment.

**Collect baseline data at similar temporal and spatial resolution.** Baseline data is critical for assessing in-water additionality, and becomes increasingly important when signals approach baseline variability and become difficult to detect and attribute to the intervention. We discourage the use of mean ocean state baselining, and encourage a dynamic baselining approach, especially for small scale

experiments. We recommend that data should be collected in baseline conditions at similar temporal and spatial resolution compared to measurements of the intervention. These data will result in a dynamic baseline that can be successfully employed to estimate additionality of the intervention, and ultimately place constraints on the total amount of resulting CDR. Improvements upon in-water baselining techniques should be prioritized for future work.

**In-water MRV may need an "intervention-only" scenario in addition to a baseline.** Comparisons directly to baseline data are applicable for TA enhancement and water tracing, but the parsing of an OAE signal becomes challenging when gas exchange and dilution are occurring simultaneously. The actual CDR signal for $f$CO$_2$ and pH is calculated through differencing a "ΔOAE" scenario, in which alkalinity is dispersed without any gas exchange, from a "ΔCDR+ΔOAE" scenario, in which excess alkalinity is allowed to take up CO$_2$ from the atmosphere (Figs. 1, 11). It is likely that signals in DIC alone, which can be directly referenced to the baseline scenario, will be too small to measure in the field. In this experiment, we calculated this dilution-only "ΔOAE" scenario and ran a model for gas exchange to calculate CDR. In co-dispersals of water tracer and alkalinity, measurements will directly reflect the "ΔOAE+ΔCDR" scenario. The "ΔOAE" scenario can be estimated through a combination of baseline carbonate chemistry data outside of the patch, and water tracer and carbonate chemistry data within the patch.

**Modeling efforts should occur in parallel.** We suggest that models be developed alongside in-water activities, to maximize the information extracted from the experiments, and to continue to refine the models used for mCDR. Focus should be put on critical measurements that can help to ground the models currently used for CDR quantification (Isometric, 2024). This includes very near-field models of ship-wake turbulence and dilution, as well as larger-scale regional ocean models of how water circulates and how CO$_2$ exchanges with the atmosphere. Integrating these multi-scale approaches will be essential for MRV by connecting local alkalinity changes tracked through ship-wake modeling with regional CO$_2$ uptake and biogeochemical patterns captured by regional modeling. These regional models should be validated both in terms of relevant baseline variability, and also in terms of their ability to capture in-water OAE and CDR perturbations, assessed via direct measurements. Ultimately, alongside field deployments, these models will serve as critical MRV tools for optimizing OAE deployment strategies while ensuring verifiable CO$_2$ removal.

## 5    Conclusions

At this early stage in OAE research and development, field experiments are important to establish limits of detection, signal to noise, and variability, and map those onto the ability to conduct environmental monitoring of experimental OAE interventions. Furthermore, tracer experiments alone, without any manipulation of carbonate chemistry, allow for comparisons to model results, which can help to identify places where in-water measurements and models agree and places where they do not.

Our successful dispersal and subsequent monitoring of an RT patch, along with a suite of sensors and measurement platforms geared towards establishing an in-water MRV framework, demonstrates that such monitoring is possible with existing instrumentation and technology. Furthermore, RT appears to be well suited for small-scale, open-ocean deployments. It is especially

beneficial given its visual identification during and for a short time after the dispersal, simplifying
operations and tracking using a variety of sensing platforms.

Based on the results of this experiment, we suggest that attention is given to sampling baseline conditions along with the intervention itself, at similar temporal and spatial resolution, collecting and measuring samples at as high resolution as possible. Pairing measurements on a variety of platforms is beneficial to combine Lagrangian and Cartesian reference frames. In addition, matching the spatial and
990 temporal scale of well-calibrated models to the spatial and temporal timescale of the in-water experiment is highly recommended.

## 6    Author Contributions

Conceptualization: AVS, JER, APMM, ZAW, DCM, KC, HHK.
Data curation: AVS, MH, LM, JER, MBG, KM.
Formal analysis: AVS
Funding acquisition: AVS, JER, APMM, ZAW, DCM, KC, HHK.
Investigation: AVS, JER, ZAW, MH, LM, CLD, FE, KM.
Methodology: AVS, JER, APMM, ZAW, DCM, MBG, HHK.
Project administration:  AVS, JER, APMM, ZAW, DCM, KC, HHK.
Resources: All coauthors
Software: AVS, JER
Supervision: AVS, JER, APMM, ZAW, DCM.
Visualization: AVS, JER
Writing – original draft: AVS
Writing – review & editing: All coauthors

## 7    Data Availability

All data – ship underway, ship bottle, CTD sensor, and drifter datasets, are publicly available on the
NOAA National Centers for Environmental Information Database:
https://www.ncei.noaa.gov/access/metadata/landing-page/bin/iso?id=gov.noaa.nodc:0305856
The data package is citable as Subhas et al. (2025).

## 8    Competing Interests

The authors declare no competing interests.

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
