# Peer review of "A tracer study for the development of in-water monitoring, reporting, and verification (MRV) of ship-based ocean alkalinity enhancement"

_EGUsphere, 2025_

## Referee Comment (RC1)

Review of Subhas et al. "A tracer study for the development of in-water monitoring, reporting, and verification (MRV) of ship-based ocean alkalinity enhancement"

This is a thoughtful and well-conducted study which attempts to take the challenge of quantifying the effects of marine carbon dioxide removal (mCDR) from the realm of theory, where it has mostly been confined, out into the real ocean. The authors convincingly use a tracer approach to simulate the addition of alkalinity to Atlantic waters and monitor the dispersal of the tracer over multiple days.

MAJOR COMMENTS

-The paper proposes an MRV framework, but it's a little unclear what the framework is, exactly. Is it Equations 1a-1d? Is it the conceptual diagrams in Figure 1? Is it all the text in Lines 90-125? Is it all of these combined? I may have missed it, but after reading the manuscript I would struggle to describe the framework to someone else in a sentence or two. It would be good to clarify the basic points of the framework, maybe in some bullet points or a callout box.

-One major question I am left with regarding the dispersal of the tracer is the path of the patch and its north-south extent. Figure 5 shows that the team followed the tracer in generally east-west transects, which slowly moved from north to south over the experiment as the tracer was advected. But did the tracer move north to south as a fairly coherent patch, with the total surface area not changing much, or did it 'smear', so the tracer patch grew in total surface area over time? Essentially, it seems to me that the patch was well-sampled in one direction (east-west), but lightly-sampled in the north-south direction due to understandable time restrictions. Figure 3b shows a lot of heterogeneity in the patch concentrations: how can the authors know that they were always sampling 'peak' patch RT concentrations in later Figures (i.e. Figures 6, 10). This seems to have large implications for the estimate of CDR efficiency starting at Line 621, since the efficiency calculation is predicated on all the added TA remaining in the patch (i.e. no precipitation). Perhaps this is mostly unknowable, but Figure 3a shows that some drone imagery was taken, and perhaps maybe satellite imagery could be available as well, which might inform this question of the patch extent.

MINOR COMMENTS

-L51: "Research"

-L562-63: I don't think semicolons are the right punctuation here.

-L74: I was not familiar with the concept of ship wake dilution models. A sentence quickly explaining what these are and what they are used for might be helpful.

-L84: fCO2 has not yet been defined

-Figure 1 and caption: most of the paper uses fCO2, not pCO2. The difference is minor, but consistency would be good.

-Figure 1: these plots show what would happen to the carbonate system in the surface ocean with infinite dilution, I think. My understanding is that for a fixed volume of seawater, OAE eventually results in a DIC that returns to near-baseline levels, but at a slightly higher concentration, and that pH also remains just slightly elevated above baseline levels. It's a little hard to tell from Fig 1, but I think DIC and pH return to baseline conditions. This is a subtle difference that might be worth mentioning.

-L109-110: Is it worth keeping the ΔTA(CDR) term in Equation 1a if it is always zero?

-L141-142: does storing CO2 in biomass require a sustained, measurable fCO2 gradient?

-L153: "(90-foot)"

-L163: strange citation format with all four authors listed

-L190: define "UHMW"

-L193: refer reader to Figure 2.

-L200: 38 knots is very fast for the *Connecticut*, unless it is a cigarette boat

-Equations 2 and 3: I'm curious if these are constructed to apply in straight-line travel conditions. How does the spiral dispersal pattern used in this experiment work in these models? It seemed like the swaths of RT sometimes dispersed into each other, which I imagine would have an effect.

-L217, it might be useful here to describe what the "in-patch"/"out of patch" thresholds were. It's later mentioned on L328.

L240-241: include metric units here (inches and feet used)

-L239: the TA was distributed at the surface, but the underway intake was at 1.5. Could this have affected anything in terms of peak RT (and thus estimated TA) concentration, patch edge detection, etc?

-L245: It seems the total time from the ship's hull to the fluorometer was about 30 seconds (mostly delayed in the debubbler). Does that lag time seem about right?

-L260: "analyzed"

_L293: "what does "subsequently routinely" mean? Were all samples not poisoned?

-L294: the 13-C DIC data are only presented in the supplementary, and not discussed anywhere that I can see. These data and associated methods probably don't need to be included. If they are needed for some reason, how was the 13-C isotope ratio calibration done (Dickson CRM is not certified for 13-C).

-L317: were the four drifters all deployed in the same place, at the same time?

-L339: what concentration of liquid NaOH is assumed?

-L345-349: this section shifts into present tense.

-L423-424: this gets back to the convening swaths of RT dispersal. It would be good to discuss the implications of this some more, in terms of OAE.

-L439: the decrease in RT concentration is hard to see with the color scale used in Fig. 5a. Also can mention here the decrease is seen in Fig. 6a as well.

-L460: It's hard to parse the three exceptions noted here. Perhaps number or identify them individually somehow.

-Figure 6: Seelmann et al. (2019 p. 526) note the occurrence of outliers in underway TA data from the Contros HydroFIA instrument. Just going on looks, the high TA points in Figure 6d of this manuscript seem to match those criteria. The Seelmann paper cites a method to filter out these outliers as well.

-Figure 7: the salinity 31.8 contours are very faint, and will be hard to see in the final paper

-L501: Is "interpolation" the right word here? I usually think of interpolation as filling in gaps in lower-frequency data to match higher-frequency measurements. As in, the TA data in this experiment could be linearly interpolated to match the higher-frequency T, S etc. Were the higher-frequency data subsampled, or averaged somehow, to match the 10-minute TA data?

-L508: describe the statistical test used to compare the fCO2 data categories.

-L515:516: describe the statistical tests and coefficients which indicate correlation between fCO2 and TA.

-L530: should be Fig 3c.

-L555 same question regarding interpolation

-Figure 9: it is really hard to tell the two boxes/sets of horizontal lines apart. The open box and shaded box are easy to mix up. One option is to use dashed horizontal lines for the mean and +/- standard deviation of TA_bl, and solid lines for mean +/- standard deviation for TA_bl,interp, or some other similar scheme. The shading doesn't really add anything for me.

-L578-587: I thought this section was really well done.

-L595, fCO2 returns to near-baseline values

-Figure 10: do all the points in this Figu correspond to points in Figures 8 or 9? Are they all matched up to the underway TA measurements?

-L606: should be uatm, not ppm.

-Figure 10 caption, Line 2: should reference Figure 8 I think, not Figure 7. Also, do you mean "RT concentration" instead of "RT signal"?

-L612: what is a "small but mean" offset?

-L623: should reference Figure 10h. Line 593 days the TA enhancement was >20 umol/kg)

-L624: should reference Figure 10e

-L627-630: Isn't this a big deal? How do the authors suggest the spatial heterogeneity of the patch should be addressed?

-L627: this focused on the peak RT concentration measured in the patch. Could it not have been higher in another spot? Wasn't the patch quite patchy (i.e. Figure 3b)?

-L634: how likely is it for the water mass to remain at the surface? The mass in this experiment advected quite a bit, and presumably mixed as well.

-L643-644: this returns to my question of patch behavior. Did the whole mass travel south, or did it spread out south with some concentration remaining at the original northerly dispersal location?

-L678: "may advect may disperse"

-L705-710: I thought this section was well done.

-Final thought: I suspect that some will read this work, see the 8% CDR efficiency number, and be disheartened for the prospect of OAE. Perhaps the authors can speculate about this some more. How much longer, or how much more broadly spatially, would

observations need to be carried out to observe a higher efficiency?  What might be the highest detectable efficiency, given uncertainties in the baseline estimates and analytical instrumentation? More broadly, what even represents a successful CDR?  One that is measureable?  One that is profitable? Presumably the CDR effects would have continued after the monitoring efforts in this experiment were done, but how might someone realistically do MRV for these ongoing $CO_2$ removals?

---

## Author Response (AR1)

**Anonymous Review 3**
**Review of 'A tracer study for the development of in-water monitoring, reporting, and verification (MRV) of ship-based ocean alkalinity enhancement' by Subhas et al.**
The work by Subhas et al. examines the results of a Rhodamine tracer release study conducted near Martha's Vineyard on the east coast of the United States. The resulting distributions of Rhodamine are used to project the effects of a similar addition of NaOH on carbonate chemistry. This hypothetical alkalinity release experiment is predicted to result in carbon dioxide removal. The ultimate goal of the study is to evaluate the use of the Rhodamine tracer release technique for MRV of ship-based ocean alkalinity enhancement. There is no doubt that this research makes a significant contribution to the growing body of literature on the feasibility of OAE as a marine carbon dioxide removal (mCDR) method. The study is well-designed, the quality of the data and the processing steps are well-documented, and both the data and Matlab code are openly shared and available to readers. In my opinion, this work is worthy of publication after addressing some limitations of the proposed methodology.

We appreciate the Reviewer's comments and provide some answers and discussion below. Several reviewers raised the issue of OAE and its MRV "research" vs. "deployment", and we think the distinction here is an important one to make. We now make this clear by referring to an OAE "analytical framework" rather than an MRV framework, and specify that this analytical framework can and should be used for the development of MRV.

My first comment concerns the calculation of dilution factors. The authors define the dilution factor of Rhodamine using a simple equation (L195), where $D = RT\_in / RT\_underway$. This approach does not appear to explicitly account for the diffusion of Rhodamine out of the patch. If the goal is to examine the decay of Rhodamine over time, such a simplification may be sufficient. However, if dilution factors produced in this way are used to estimate the decay of added NaOH, the simplification may lead to inaccurate results. Therefore, it is crucial not to use empirically derived parameterizations without considering the underlying processes.

Dilution is only used in the initial stages during the release itself (approximately 90 minutes), and is used to compare with the semi-empirical ship-wake model of Chou (1996). This equation simply calculates the measured concentration in relation to the initial concentration, and the reviewer is correct that it does not distinguish between how that dilution occurs, i.e. by advection, lateral diffusion, or vertical mixing. Thus, this dilution calculation aligns with the immediate turbulence impact on tracers, similar to how ship wake models parameterize this effect. We plan on adding a short introduction and description of ship-wake models for turbulent dilution in the Methods, and a few sentences summarizing the below content in Section 4.1 of the manuscript.

In the case of dilution, Rhodamine will behave similarly to NaOH. Diffusion, however, depends on the gradient (TA or Rhodamine) between the patch and the surrounding water and could drive NaOH both into and out of the patch, whereas Rhodamine will always

diffuse outward, as it is not a naturally occurring tracer. I would argue that diffusion may play a more significant role—indeed, the decay of Rhodamine in Fig. 6 appears clearly exponential, even on a log scale. While I recognize that the measured range of TA is not large enough to produce drastically different outcomes, it remains important to correctly parameterize the dispersion model. The use of the dynamic baseline seems to improve the results, suggesting that even small variability in TA matters.

As mentioned above, dilution of RWT (and of alkalinity added to seawater) will occur via multiple processes. In all cases, these added materials can only diffuse out of the patch – whereas background constituents and other components of seawater could be exchanged bi-directionally. However, given the timescales and spatial scales above, it is important to distinguish physical mixing and stirring processes, which are often characterized using eddy diffusivities, and will operate on entire water parcels, versus molecular diffusion, which happens on much smaller spatial scales (and much longer timescales). For instance, lateral eddy diffusivity is on the order of ~5 m$^2$/s (Rypina et al., 2019). Scale analysis of $x^2/t \sim 5$, operating on a patch ~500m in scale such as the one in our experiment, gives a characteristic eddy diffusion timescale t of $500^2/5 = 50,000$ seconds, or 13 hours. Thus, while not important initially, this process clearly controls the mixing and tracer loss on timescales of days. Vertical diffusivity is significantly slower, order $10^{-4} – 10^{-5}$ m$^2$/s, and molecular diffusion is orders of magnitude slower at ~$10^{-9}$ m$^2$/s. We note again, however, that these mixing processes operate on entire water parcels, not on individual components, and it is our hypothesis that eddy-driven mixing and stirring will dilute all components of the patch in a similar manner. It will be critical to carefully evaluate the distribution of RT and NaOH in paired release experiments to test this hypothesis; it will also be useful to confirm that this behavior is captured in ocean models as well. This is our plan for the next field experiment, and for future modeling efforts. We added the following sentences in Section 4.1:

"Ship-wake dilution models explicitly separate near-field from further dilution regimes (Chou, 1996), and focus on empirical dilution in the near-field (e.g. the ratio between measured and initial concentration; eq. 2) rather than attributing changes in concentration to specific processes such as eddy diffusivity or advective mixing. Given the turbulent nature of near-field mixing, we anticipate that almost all of the mixing was driven by the ship wake, and not by horizontal diffusivity (with a lateral eddy diffusivity of ~5 m$^2$ s$^{-1}$, Rypina et al., 2019); vertical diffusivity (~$10^{-4}$ m$^2$ s$^{-1}$, Rypina et al., 2019) or molecular diffusion (~$10^{-9}$ m$^2$ s$^{-1}$, Zeebe, 2011)."

Related to the previous comment, could the authors discuss how the results of the study apply to the use of solid alkalinity sources (e.g., lime, Mg(OH)$_2$)? I imagine that the proposed framework could be applicable to future deployments, and it is important to understand the limitations of the method.

The advantage of liquid alkalinity is that it can be tracked similarly to the RT dye, whereas solid alkalinity sources pose additional monitoring challenges. Solid -based OAE efficiency

will depend on how fast the material dissolves and sinks out of the mixed layer, and both of these processes will decouple the resulting alkalinity enhancement from the dye tracer. We will add these additional considerations – dissolution rate and sinking of solid alkalinity -- in the paper, both in setting up the OAE analytical framework, and in the conclusions. We added the following sentence to section 2.1:

"In addition, solid feedstock dissolution rate and sinking velocity would need to be considered, and would spatially and temporally decouple alkalinity generation from $CO_2$ uptake at the sea surface. "

And in Section 4.5:

"In the case of solid feedstocks, vertical sampling becomes critical as the interaction between sinking velocity and dissolution rate will produce a profile of alkalinity generation throughout the water column (Feely et al., 2002, Sulpis et al., 2021, Subhas et al., 2022)."

Finally, I agree with the authors on the benefits of using the dynamic baseline. However, in a larger-scale deployment, identifying or defining "out-of-patch" conditions could be challenging—especially in the absence of a visible tracer. Moreover, once alkalinity spreads across large areas with varying oceanic conditions and water masses, the concept of "out-of-patch" may lose its relevance. It may be necessary to sample along the patch boundaries, gather several baselines, and integrate them into the proposed framework. A stronger recommendation—which the authors do emphasize in their conclusions—would be to constrain the baseline prior to the experiments by constructing better empirical models (requiring high-resolution data over at least an annual cycle) and characterizing the sources of baseline variability, which could later be incorporated into model-derived counterfactuals.

Thanks for this – we do believe that our sampling plan effectively accomplished this "along-boundary" approach, because we were constantly passing into and out of the patch, thus sampling the boundary multiple times. Indeed, this approach is what allowed us to construct the dynamic baseline and demonstrate that we were capturing "true" spatial and temporal variability along with any signal of the perturbation itself. We can strengthen this "boundary sampling" approach to the recommendations, as well as understanding how these boundaries might interact with interannual variability.

We added:
"Baseline data is critical for assessing in-water additionality, and becomes increasingly important when signals approach baseline variability and become difficult to detect and attribute to the intervention."
And in the modeling section:
"These regional models should be validated both in terms of relevant baseline variability, and also in terms of their ability to capture in-water OAE and CDR perturbations, assessed via direct measurements."

**Minor typos and comments:**

- **L51:** Typo – "Research" (please correct). Done.

- **L284:** Please verify the sensor models—something seems incorrect. Oxygen is likely SBE 43; please confirm the same for CTD. Confirmed – thank you for catching this, we had simply duplicated the pH sensor for oxygen. Changed Oxygen to SBE43.

- **L359:** Please use a consistent reference format for Guo et al. throughout the manuscript (e.g., "in review" or "in revision"), unless the paper has already been accepted. Thank you -- This paper has now been accepted for publication and our citation is revised accordingly. We anticipate that the article will be fully published by the time this manuscript is deemed suitable for publication.

**Anonymous Review 1**

Review of Subhas et al. "A tracer study for the development of in-water monitoring, reporting, and verification (MRV) of ship-based ocean alkalinity enhancement"

This is a thoughtful and well-conducted study which attempts to take the challenge of quantifying the effects of marine carbon dioxide removal (mCDR) from the realm of theory, where it has mostly been confined, out into the real ocean. The authors convincingly use a tracer approach to simulate the addition of alkalinity to Atlantic waters and monitor the dispersal of the tracer over multiple days.

We thank the Reviewer for their constructive thoughts and comments on our manuscript. We provide detailed responses below.

MAJOR COMMENTS

-The paper proposes an MRV framework, but it's a little unclear what the framework is, exactly. Is it Equations 1a-1d? Is it the conceptual diagrams in Figure 1? Is it all the text in Lines 90-125? Is it all of these combined? I may have missed it, but after reading the manuscript I would struggle to describe the framework to someone else in a sentence or two. It would be good to clarify the basic points of the framework, maybe in some bullet points or a callout box.

Thank you for this point, we will clarify this in the appropriate section of the manuscript, and may shift our language to discussing it as our "OAE analytical framework" rather than as an actual MRV approach. We appreciate the suggestion of a callout box, and will consider this, or a succinct set of bullets in the framework section, to make it clear

what we see as the important steps. To us, this focuses on the following statement made in the manuscript:

"We explicitly distinguish between OAE-driven signals and CDR-driven signals."

The analytical framework then follows as three main steps, which we now outline in the beginning of Section 2.1:

"Constructing an analytical framework for mCDR, and for OAE specifically, requires defining the main processes at work. Here we define three main steps in our analytical framework:

1. Net Alkalinity transfer from alkaline feedstock into seawater via dissolution;
2. Tracking of dissolved alkalinity and its dispersion (and for solid feedstocks, particle transport, dissolution, and settling);
3. Calculation of CDR due to the above processes, via direct measurement, models, and/or a combination of both.

Here, we assume a liquid form of alkalinity e.g. sodium hydroxide, such that in Step 1 alkalinity transfer efficiency is very high and can be restricted to the sea surface. In Step 2, we solely focus on dissolved alkalinity tracking.

For all steps, a baseline state must be established. ..."

-One major question I am left with regarding the dispersal of the tracer is the path of the patch and its north-south extent. Figure 5 shows that the team followed the tracer in generally east-west transects, which slowly moved from north to south over the experiment as the tracer was advected. But did the tracer move north to south as a fairly coherent patch, with the total surface area not changing much, or did it 'smear', so the tracer patch grew in total surface area over time? Essentially, it seems to me that the patch was well-sampled in one direction (east-west), but lightly-sampled in the north-south direction due to understandable time restrictions. Figure 3b shows a lot of heterogeneity in the patch concentrations: how can the authors know that they were always sampling 'peak' patch RT concentrations in later Figures (i.e. Figures 6, 10). This seems to have large implications for the estimate of CDR efficiency starting at Line 621, since the efficiency calculation is predicated on all the added TA remaining in the patch (i.e. no precipitation). Perhaps this is mostly unknowable, but Figure 3a shows that some drone imagery was taken, and perhaps maybe satellite imagery could be available as well, which might inform this question of the patch extent.

We now include satellite imagery (see below) to confirm that the patch started to smear east-west, and that we sampled through the longest part of it. While we can never be

absolutely sure that we sampled the peak, the constancy and decay of the peak RT signal in the underway system suggests that we were not randomly sampling the patch but instead systematically passing through the peak concentration every time.

In all three images, the patch can be clearly visualized, and demonstrates that the patch did stay coherent and move as a unit, rather than getting dispersed irregularly. Over time it stretches mostly east-west, with some southwest-northeast trending observed at 24 hours. This matches well with the long axis of sampling demonstrated in fig. 5. We plan on adding context to Figs. 5 and 6 to indicate where and when these images were taken with respect to the spatial ship track and underway timeseries data streams.

[Figure]

Proposed figure addition: Satellite imagery collected via Planet Labs at 3 time points: a) approximately 1 hour after dispersal; b) approximately 6 hours after dispersal; and c) approximately 24 hours after dispersal. Images shown are true color, scaled to enhance image brightness. Scale bars in each image are 1km.

Methods addition: High resolution satellite imagery was collected during the cruise from Planet Labs via two methods: 1) Ultra-high resolution (0.5m per pixel) multiband imagery (red, green, blue, near-infrared, and panchromatic) was collected via the Planet SkySat constellation through tasked image collection and 2) high resolution (3.0m per pixel) multispectral imagery (8-band) through the PlanetScope near-daily revisit product via the Dove/SuperDove constellation. Three images were collected during the cruise, on September 2, 2023 at 14:14:25 UTC, September 2, 2023 at 18:58:16 UTC, and September 3, 2023 at 14:43:07 UTC. The first two images were collected via SkySat tasking, and the third via PlanetScope. Level 0 images were internally processed by Planet's algorithms for orthorectification and atmospheric correction to produce Level 3 surface reflectance data. Orthorectification was verified using shipboard location data, which required small corrections for images two and three.

[Figure]

Updated underway data map showing satellite image capture windows in panel a.

[Figure]

Updated underway timeseries figure showing satellite image collection times in purple triangles along the x-axes.

MINOR COMMENTS

-L51: "Research"  Fixed, thank you.

-L562-63: I don't think semicolons are the right punctuation here. Revised accordingly.

-L74: I was not familiar with the concept of ship wake dilution models. A sentence quickly explaining what these are and what they are used for might be helpful. Thank you, we will be sure to add some context here.

-L84: fCO2 has not yet been defined Thank you, we will add a definition of fCO2.

-Figure 1 and caption: most of the paper uses fCO2, not pCO2. The difference is minor, but consistency would be good. Thank you for the catch, we have now changed to fCO2 throughout.

-Figure 1: these plots show what would happen to the carbonate system in the surface ocean with infinite dilution, I think. My understanding is that for a fixed volume of seawater, OAE eventually results in a DIC that returns to near-baseline levels, but at a slightly higher concentration, and that pH also remains just slightly elevated above baseline levels. It's a little hard to tell from Fig 1, but I think DIC and pH return to baseline conditions. This is a subtle difference that might be worth mentioning.  Thank you, yes we assume infinite dilution, i.e. the case where the intervention is relatively small relative to the volume of the body of water. Interventions would have to become significantly larger (gigatonne-scale), or the body of water significantly more restricted, for there to be a steady-state mean shift in TA, DIC, or other carbonate system parameters. We now make this clear.

-L109-110: Is it worth keeping the ΔTA(CDR) term in Equation 1a if it is always zero? We prefer to keep it in for completeness, as it is unclear for now how feedbacks associated with CDR on the TA signal will be made and assigned.

-L141-142: does storing CO2 in biomass require a sustained, measurable fCO2 gradient? This is an interesting point. Largely, the answer is no, because some groups/companies are considering terrestrial biomass waste storage in the deep ocean, which would not generate a CO2 gradient in seawater during the biomass formation. But if the biomass is grown in seawater, measuring the decrease in the fCO2 in surface water as a result of that photosynthetic activity would help with its MRV.

-L153: "(90-foot)" Thank you, changed.

-L163: strange citation format with all four authors listed Thank you, fixed.

-L190: define "UHMW" Thank you, we just decided to remove that acronym for clarity as it did not add much.

-L193: refer reader to Figure 2. We think the best figure to reference here would be Fig. 3, which shows the image of the patch. We do so, but note that this does now reference figures out of order in the text.

-L200: 38 knots is very fast for the *Connecticut*, unless it is a cigarette boat Thank you for the catch! It should be 3.8 knots, changed.

-Equations 2 and 3: I'm curious if these are constructed to apply in straight-line travel conditions. How does the spiral dispersal pattern used in this experiment work in these models? It seemed like the swaths of RT sometimes dispersed into each other, which I imagine would have an effect.  This is an interesting point. These equations were designed to operate on a ship traveling in a straight line, discharging waste or ballast behind it. At least with the first pass over the patch, the underway system intake is always upstream of the RT discharge, so we would only "see" the original material on the first pass-through.  On later passes, though, there would be some cumulative effect and the dilution would actually be less than observed. We now make a note of this, and indicate that this approach provides an upper bound on dilution, which we may not have reached by dispersing in this way. We added in section 4.1:

"This model also assumes that discharge occurs behind a vessel moving in a straight line, rather than in a spiral pattern with overlapping layers of dispersed material."

-L217, it might be useful here to describe what the "in-patch"/"out of patch" thresholds were. It's later mentioned on L328. Thank you, added here for clarity:

"We attempted to conduct CTD casts at the highest RT concentrations measured on the underway system, although ship drift meant that we did not hit the peak once the CTD entered the water. Out-of-patch locations were determined visually by reaching near-baseline underway RT concentrations.  "

L240-241: include metric units here (inches and feet used) Thank you, added.

-L239: the TA was distributed at the surface, but the underway intake was at 1.5. Could this have affected anything in terms of peak RT (and thus estimated TA) concentration, patch edge detection, etc? We don't think so, as the dye appeared relatively well-mixed in the upper portion of the ocean, as shown by Fig. 7 and S1. CTD sensor readings less than 1-2m using the CTD rosette are very difficult to interpret, regardless, because of ship heave.

-L245: It seems the total time from the ship's hull to the fluorometer was about 30 seconds (mostly delayed in the debubbler). Does that lag time seem about right? Yes, that is about

what we calculated as well. There was a typo that said "3" seconds when it should have been "30. This is now fixed.

-L260: "analyzed" Thank you, fixed.

_L293: "what does "subsequently routinely" mean? Were all samples not poisoned? Edited for clarity. All samples were poisoned.

-L294: the 13-C DIC data are only presented in the supplementary, and not discussed anywhere that I can see. These data and associated methods probably don't need to be included. If they are needed for some reason, how was the 13-C isotope ratio calibration done (Dickson CRM is not certified for 13-C). We provide these data because we collected them, and think it is important to include here for completeness. But agreed that we did not use them in the paper. We provide some details on our calibration procedure for d13C in the methods now:

"Absolute $\delta^{13}$C for seawater standards was calibrated by running seawater against solid reference materials (e.g. IAEA-C2, NBS-19, NBS-18) on an Automate prep device coupled to the same Picarro G-2131i (Subhas et al., 2015, Subhas et al., 2019)."

-L317: were the four drifters all deployed in the same place, at the same time? Yes, they were deployed essentially all at once, at the same location. We were interested to see how far they would drift from each other, which the traces demonstrate is minimal. We add some clarification and some discussion on this now.

"All four drifters were deployed at once, at the same location, to assess how much they would drift from eeach other, and the patch, over the deployment."

-L339: what concentration of liquid NaOH is assumed? We assume the addition of 50% by weight NaOH (19.1 molar).

-L345-349: this section shifts into present tense. Thank you, edited for consistency.

-L423-424: this gets back to the convening swaths of RT dispersal. It would be good to discuss the implications of this some more, in terms of OAE. Thank you, we add a sentence of two of discussion here on this point as well. This may influence the small scale "patchiness" of the patch, and will not be captured in larger-scale models, which will assume a uniformly distributed patch of OAE. Something similar could potentially happen with coastal outfalls, given tides, currents, and wind that could "pile" up water on top of itself. Therefore, it is likely critical to investigate these very near-field processes in some more detail, with both models and observations.

-L439: the decrease in RT concentration is hard to see with the color scale used in Fig. 5a. Also can mention here the decrease is seen in Fig. 6a as well. Thank you, added this for ease of reading/comprehension.

-L460: It's hard to parse the three exceptions noted here. Perhaps number or identify them individually somehow. Thank you, we changed the wording to make this section a bit clearer.

-Figure 6: Seelmann et al. (2019 p. 526) note the occurrence of outliers in underway TA data from the Contros HydroFIA instrument. Just going on looks, the high TA points in Figure 6d of this manuscript seem to match those criteria. The Seelmann paper cites a method to filter out these outliers as well. Thank you for this – we did filter outliers, but did it by eye. The Seelmann et al. (2019) approach was for a single seawater sample measured over time, and therefore the Grubbs outlier test they used was sufficient. However, because we sample spatial and temporal gradients, this univariate approach is not appropriate. We have tried a few different methods of removing outliers mostly focusing on calculating a running mean of 5-10 samples (corresponding to 40 minutes to 80 minutes in window width) and finding outliers either through a standard deviation cutoff or a z-test ($p < 0.05$) cutoff. We found similar results with these statistical methods and our by-eye method, all resulting in about 2-3% of the data points being removed. We will discuss the approach briefly in the paper, and can include a figure in the supplemental identifying the outliers removed.

-Figure 7: the salinity 31.8 contours are very faint, and will be hard to see in the final paper Thank you, we made them stronger in this latest version:

[Figure]

-L501: Is "interpolation" the right word here? I usually think of interpolation as filling in gaps in lower-frequency data to match higher-frequency measurements. As in, the TA data in this experiment could be linearly interpolated to match the higher-frequency T, S etc. Were the higher-frequency data subsampled, or averaged somehow, to match the 10-minute TA data? Thanks for this – yes, the appropriate word may be "downsampled", as we downsampled the T,S, and fCO2 data, matching the closest timestamped measurement to the TA measurement timestamp.

-L508: describe the statistical test used to compare the fCO2 data categories. we used a two-way t-test and now provide p values for the comparison. For fCO2, the categories are the same (p=0.4), and similarly for TA (p=0.15).

-L515:516: describe the statistical tests and coefficients which indicate correlation between fCO2 and TA. We used a linear model ("fitlm" in MATLAB) and now provide those as supplemental figures, along with the goodness of fits (r-squared). Another reviewer also wanted to see salinity-and temperature normalized TA and fCO2, which we also include in this supplemental figure as panels c and d.

[Figure]

-L530: should be Fig 3c. Thank you, edited.

-L555 same question regarding interpolation Changed to clarify and be more specific.

-Figure 9: it is really hard to tell the two boxes/sets of horizontal lines apart. The open box and shaded box are easy to mix up. One option is to use dashed horizontal lines for the mean and +/- standard deviation of TA_bl, and solid lines for mean +/- standard deviation for TA_bl,interp, or some other similar scheme. The shading doesn't really add anything for me. Thank you, we have edited this figure to add colors for clarity.

-L578-587: I thought this section was really well done. Thank you.

-L595, fCO2 returns to near-baseline values Edited

-Figure 10: do all the points in this Figu correspond to points in Figures 8 or 9? Are they all matched up to the underway TA measurements? These are now the peak values in each hourly bin, so are a subsample of the values plotted in Figures 8 and 9. Figure 8 shows all TA and fCO2 data, downsampled to match the TA sampling resolution. Figures 9 and 10 only show the peak hourly data. we clarify this now in the text.

-L606: should be uatm, not ppm. Fixed, thank you.

-Figure 10 caption, Line 2: should reference Figure 8 I think, not Figure 7. Also, do you mean "RT concentration" instead of "RT signal"? Fixed this, thank you

-L612: what is a "small but mean" offset? We mean that there is a small but statictically insignificant offset between the mean values, edited for clarity.

-L623: should reference Figure 10h. Line 593 days the TA enhancement was >20 umol/kg) Thank you, changed line 593 to "just under 20 umol/kg".

-L624: should reference Figure 10e Added

-L627-630: Isn't this a big deal? How do the authors suggest the spatial heterogeneity of the patch should be addressed? This is a good question. One approach would be to try to capture the tracer distribution with imagery, as we have done with satellites. Another method would be to apply statistical and/or gas exchange models to the RT (and OAE) distribution over time, but one would somehow need to know the representative area associated with each concentration measurement. This could be accomplished using drifting assets equipped with sensors, sampling a larger area of the plume continuously over time. This spatial heterogeneity is what makes MRV so difficult using measurements, and yet these measurements are critical to provide some level of validation to models being used to calculate CDR.

-L627: this focused on the peak RT concentration measured in the patch. Could it not have been higher in another spot? Wasn't the patch quite patchy (i.e. Figure 3b)? This is true, but we do think we were sampling the peak concentration consistently, as described above.

-L634: how likely is it for the water mass to remain at the surface? The mass in this experiment advected quite a bit, and presumably mixed as well. This is a good point. It really depends on the conditions, and whether there are other water masses in the area that could interact with the patch. For instance, in coastal areas with significant freshwater input, seawater will readily subduct under freshwater lenses. So, this study is likely more appropriate for open-ocean scenarios where the water column is better stratified and there is less horizontal heterogeneity. It also clearly mixed, as the concentration decreased

steadily over time. However, interestingly, it did stay as a coherent "unit", meaning that these features can be tracked over time with precision.

-L643-644: this returns to my question of patch behavior. Did the whole mass travel south, or did it spread out south with some concentration remaining at the original northerly dispersal location? The patch did move as a coherent unit, and while it did, it spread and diluted with surrounding water. We hope this is made clear with the satellite images.

-L678: "may advect may disperse" Fixed, thank you.

-L705-710: I thought this section was well done. We appreciate your feedback!

-Final thought: I suspect that some will read this work, see the 8% CDR efficiency number, and be disheartened for the prospect of OAE. Perhaps the authors can speculate about this some more. How much longer, or how much more broadly spatially, would observations need to be carried out to observe a higher efficiency? What might be the highest detectable efficiency, given uncertainties in the baseline estimates and analytical instrumentation? More broadly, what even represents a successful CDR? One that is measureable? One that is profitable? Presumably the CDR effects would have continued after the monitoring efforts in this experiment were done, but how might someone realistically do MRV for these ongoing CO2 removals? These are interesting, at times philosophical, but highly relevant questions for in-water measurements for the purpose of MRV of OAE. While we do not feel we can extrapolate our signals to beyond our dataset, obviously the best scenario would be to follow the patch and measure it until it completely returns to below detection limits, but even in ideal scenarios for small-scale experiments, this timescale is likely on the order of days. This would allow a more complete budgeting and assess more complete efficiency directly from measurements. We do acknowledge that at some point, models must be used to estimate CDR, because it will become unmeasurable, but measurements that can help ground -truth and validate these models are essential at this stage. We add some discussion on this topic now.

> "Although this is a relatively small portion of the total CDR, other models have predicted CDR via OAE to take months or years (Zhou et al., 2024, He and Tyka, 2023). Thus, the potential for directly observing nearly 10% of the total CDR, on the timescale of days, represents a critical opportunity to ground-truth the CDR potential of OAE through in-water campaigns."

**Anonymous review 2**

I found this paper to be extremely well written, very nicely presenting the results of the field work tracking rhodamine tracer as a preliminary step to tracking a future OAE experiment via the dispersal of a NaOH solution. I commend the authors on stepping through every

aspect of the experimental design, deployment, interpretation, and implications for the future OAE experiment in a highly readable and comprehensive way. We thank the reviewer for their kind words.

I have only one somewhat major suggestion, which doesn't require new analysis but might trigger some careful thinking about reframing the motivation and recommendation of the paper. I think my suggestion is related to Reviewer #1's question about "What is your MRV framework?" I see the work presented in the paper as foundational research that can inform an MRV framework, but not itself MRV nor a framework. The term MRV signals a set of processes used in a voluntary or regulated carbon marketplace, whereby carbon removal is monitored and reported to a third party who verifies that it met a given set of predefined standards. The language in the paper should make this distinction. We thank the reviewer (and Reviewer 1) for pointing out this key distinction, and it is one that we have now worked into the manuscript throughout – specifically in the introduction and in the recommendations at the end -- to distinguish this research effort from what industrial/practical MRV could look like. In the end, we decided to reframe our approach as an "OAE analytical framework", rather than as a specific MRV methodology, which we agree would be difficult to implement across the board.

Further, I suggest the authors to reflect on how much of this kind of extraordinary research effort (deep domain expertise, research vessel and state-of-the-art laboratory infrastructure, a valuable equipment pool, and the funding to support the team) they would recommended for future MRV. At a minimum, the target audience for the work should be made explicit (my guess: researchers trying to design a comprehensive field experiment with relevance for tracking surface anomalies influencing air-sea gas exchange). It would be slightly more ambitious, but commensurately more valuable, for the authors to reflect on a trajectory for mCDR researchers and practitioners that uses the knowledge gained from this research to ultimately create a more parsimonious MRV framework for operational mCDR (or at least provide recommendations on how to bridge that gap). Thank you – we have now added some content to the paper to address this important point. While we agree that this level of effort could, or should, be conducted by every group attempting MRV in the future, we do feel that it is important for in-depth, in-water research to accompany early field projects to underlie the continued development of in-water measurements for the purposes of MRV, and to help continue grounding models with these critical observations. Right now, it is unclear what "operational" MRV should look like, and studies like this one can help guide the field towards a meaningful, trusted MRV framework.

" We acknowledge that the approach outlined here is likely not suitable for routine, mature MRV. However, we emphasize that in-depth, in-water research is a necessary and critical component of early field projects, and should continue to be a focus of all groups attempting to quantify the CDR potential of OAE during new stages of deployment."

Minor comments:

The first line of abstract is deeply misleading: The solution to a 10^10th ton emissions problem is not meaningfully "supplemented" by processes that now collectively sum to 10^4th tons of removal. Please remove this sentence. Removed.

Line 30: Wording is a little confusing. I think you mean that "Over 36 hours, the ensuing anomalous carbon dioxide uptake by the ocean was driven by the enhanced air-sea gradient in fCO2. The calculated CDR signal was detectable as a 3 uatm surface ocean fCO2 increase, a pH decrease of 0.003 units, and...". If this is an accurate interpretation of what is written, it would be more understandable to the reader to spell it all out. Thank you, we edited this as you suggested for clarity.

51: typo: Researc Fixed.

Figure 1 schematic is very helpful and well done. Thank you, we appreciate the feedback.

Line 133 and throughout - specify air-sea gradient in fCO2 (or name a variable $\Delta fCO_2$) for clarity thank you, done.

357 - Why not make the gas exchange coefficient wind- or wind- and wave- dependent, as we know it is sensitive to wind speed and wave height. It would seem hard to justify using a constant, given the variability in Figure 2. We now calculate an hourly gas exchange coefficient from the Block Island wind data shown in Figure 2, and use that in our calculations. Noticeably, because the winds picked up significantly over the course of the dispersal, and gas exchange velocity is a function of windspeed squared, the flux of CO2 did increase slightly relative to the mean windspeed used in the previous verions of the manuscript. Now, the mean gas transfer velocity is ~12 cm/s (vs. ~10 previously). We now plot the gas transfer velocity, calculated via the met data, in Fig. 2, and use this hourly gas transfer velocity to update our modeled CO2 uptake. The signals grow to 4 uatm surface fCO2 increase, 0.004 pH unit derease, and an increase in DIC of 1.8 umol/kg. The manuscript is changed throughout to reflect this.

Figure 2: Specify that bathymetry is in m. Zooming closer to the ship track would be helpful, as would adding a circle of the size of the initial dispersal spiral. We include another panel of this figure that shows a zoomed in view of the patch and the dispersal

location itself. We also include CO2 gas exchange velocity as a panel, now that we use it in the manuscript for the flux calculations.

[Figure]

Figure 3: If you had a drone photo of the patch taken at a later time than Figure 3a that showed the stretching and spreading of the dye, I think it would be super helpful for the reader. We now include satellite imagery for 3 timepoints which clearly shows the patch and its spreading (and stretching), and include some discussion on this point and how it maps on to our monitoring track in Fig. 5.

[Figure]

Proposed figure addition: Satellite imagery collected via Planet Labs at 3 time points: a) approximately 1 hour after dispersal; b) approximately 6 hours after dispersal; and c) approximately 24 hours after dispersal.

In all three images, the patch can be clearly visualized, and over time stretches mostly east-west, with some southwest-northeast trending observed at 24 hours. This matches well with the long axis of sampling demonstrated in fig. 5. We have added context to Figs. 5

and 6 to indicate where and when these images were taken with respect to the ship track and underway data streams.

[Figure]

Updated underway data map showing satellite image capture windows in panel a.

[Figure]

Updated underway timeseries figure showing satellite image collection times in yellow triangles along the x-axes.

Figure 6: consider adding salinity-normalized TA to panel d, and temperature and salinity-normalized fCO2 to panel e.  The variability in T and S will be reflected in the carbon variables in a way that would be helpful to separate. We feel that salinity-normalizing fCO2 is overly complicated, but agree that temperature-normalizing fCO2 is interesting to look at, as is salinity normalized TA. We have done this now, and as expected, nTA shows no significant variability with salinity (implying that salinity drives the TA gradient between the water masses). Temp-normalized fCO2, however, shows an even larger gradient as a function of salinity – upwards of 100 uatm-- suggesting that thermodynamic effects cannot be driving these changes. Instead, it is likely the biogeochemical signature of these two water masses. We include these panels in the supplemental, rather than in the main text.

[Figure]

Figure with four panels (a, b, c, d):

Panel a: TA (µmol kg$^{-1}$) vs S. $TA = (68.8\pm12.0)S-35.6\pm380.1; r^2 = 0.15$

Panel b: fCO2 (µatm) vs S. $fCO_2 = (-466\pm32)S+15259\pm1001; r^2 = 0.54$

Panel c: nTA$_{S=35}$ (µmol kg$^{-1}$) vs S.

Panel d: $fCO_{2, T=25C}$ (µatm) vs S.

569 - DIC should also be sensitive to biology, no? This is true, but likely not on the 36-hour timescale shown here, given the magnitude of any biological changes and the size of the DIC reservoir (for the same reason that OAE-based CO2 uptake takes a long time to happen, and to see the DIC increase).

Figure 9 - I suggest using some colors! There are several solid black lines, so that it's hard to track which one the caption refers to. Thank you, we have now added colors to this plot to make it clearer.

[Figure]

Figure 10 - I'm confused why there seems to be only 1 dot per hour, when the sampling frequency was 10-minute. We discussed in the methods and above why we chose to use the peak value every hour – it makes for a more interpretable and cleaner result for this thought experiment. During real OAE deployments, data should obviously be collected and analyzed at as high of a frequency as possible, and per another reviewer's suggestion, we add some discussion about spatial heterogeneity and how to handle/measure it.

606 - yellow minus blue values — add note here (and/or in the caption) that the points are barely visible because of overlap in Figures 10 f and g Thank you, made this clarification.

628 - 638 "MRV approaches that can accurately capture the entire patch budget ... " This is related to my "major suggestion" above:  This statement makes it sound like MRV would be done in a way that mimics this experiment (with RT or similar tracer and a research vessel tracking it for days). But doing such hugely expensive field campaigns is a research-level activity that is likely beyond the scope of scalable MRV.  In fact, I'd be shocked if 1 ton of CO2 wasn't emitted by the MRV activities in their totality (4 days on the R/V Connecticut + chaser boat + all the instruments, supplies, and shipping), cancelling most/all of the mCDR here.  Thus, I see this as a research activity, rather than an MRV approach, and I am very interested in the authors' thoughts on the difference. We thank the reviewer for this important clarification, and will make sure to clearly separate research activities (such as this one) from "practical" MRV in the revised manuscript.

"We acknowledge that the approach outlined here is likely not suitable for routine, mature MRV. However, we emphasize that in-depth, in-water research is a necessary and critical

component of early field projects, and should continue to be a focus of all groups attempting to quantify the CDR potential of OAE during new stages of deployment."

---

## Author Response (AR2)

Thank you for accepting our manuscript. We checked the figures and all numbering appears to be correct now. We also changed the data availability section and added the up to date citation for the full dataset.